# Molecular basis for proofreading by the unique exonuclease domain of Family-D DNA polymerases

Leonardo Betancurt-Anzola[1,2,3,4], Markel Martínez-Carranza[1], Marc Delarue [1], Kelly M. Zatopek [2] ✉, Andrew F. Gardner[2] ✉ & Ludovic Sauguet [1] ✉

Replicative DNA polymerases duplicate entire genomes at high fidelity. This feature is shared among the three domains of life and is facilitated by their dual polymerase and exonuclease activities. Family D replicative DNA polymerases (PolD), found exclusively in Archaea, contain an unusual RNA polymerase-like catalytic core, and a unique Mre11-like proofreading active site. Here, we present cryo-EM structures of PolD trapped in a proofreading mode, revealing an unanticipated correction mechanism that extends the repertoire of protein domains known to be involved in DNA proofreading. Based on our experimental structures, mutants of PolD were designed and their contribution to mismatch bypass and exonuclease kinetics was determined. This study sheds light on the convergent evolution of structurally distinct families of DNA polymerases, and the domain acquisition and exchange mechanism that occurred during the evolution of the replisome in the three domains of life.

DNA replication is a ubiquitous process necessary for cellular growth and transfer of genetic information. This process is carried out at high fidelity by replicative DNA polymerases (repDNAPs)[1–3]. Accurate replication is dependent on the geometry of base pairing, the structure of the nitrogenous bases[4], and the selectivity of the polymerase active site[5]. In the event of a misincorporation, repDNAPs engage proofreading activity that detects and excises the misincorporated nucleotide[6]. Both correct incorporation and exonuclease activity of repDNAPs are essential for highly accurate DNA replication[7,8]. Four different families of DNA polymerases from the three domains of life have evolved to perform DNA replication: A, B, C, and D. These four families contain three distinct polymerase catalytic folds: Klenow-like (A and B-Families), Polβ-like (C-Family), and double-Ψ-β-barrel (D-Family). In Bacteria, the genome is replicated by DNA polymerases belonging to the A- and C-families[9]. In Eukarya, genomic DNA is copied mainly by three distinct repDNAPs, Polα, Polδ, and Polε, which all belong to the B-family[10]. Finally, Archaea replicate their genomes utilizing DNA polymerases from families B and D[11,12].

In addition to the three distinct polymerase catalytic folds, there are three distinct exonuclease folds: DnaQ-like, polymerase and histidinol phosphatase (PHP), and Mre11-like phosphodiesterase (PDE)[13]. Various polymerase families trapped in their exonuclease mode have been structurally determined. These include Taq Klenow fragment (A family)[14], RB69 DNA Pol (B family)[15], archaeal PolB (B family)[16], DNA Pol III (C family)[5,17], and most recently Polγ (A family)[18]. These structures, which all contain a DnaQ-like exonuclease domain, display a common mechanism of stabilizing single stranded DNA at the exonuclease active site, with the terminal phosphodiester bond properly poised within the active site. The DnaQ-like proofreading fold (Fig. 1a), which exists in all repDNAPs, except family D and a small subset of bacterial family C, is highly conserved[19]. Despite different active site architectures, all repDNAP exonuclease folds bind two $Mg^{2+}$ ions, which are required for phosphodiester bond cleavage[20,21].

Most Archaea, including the emerging Asgard superphylum[22], encode an unusual heterodimeric D-family DNA polymerase (PolD), composed of two subunits: DP1, the small subunit, contains a Mre11-like PDE fold and carries the exonuclease activity[23,24], and DP2, the large

[1]Architecture and Dynamics of Biological Macromolecules, Institut Pasteur, Université Paris Cité, CNRS, UMR 3528 Paris, France. [2]New England Biolabs Inc., 240 County Road, Ipswich, MA 01938, USA. [3]New England Biolabs France, 5 Rue Henri Auguste Desbruères, 91000 Évry-Courcouronnes, France. [4]Sorbonne Université, Collège Doctoral, ED 515, Paris, France. ✉e-mail: kzatopek@neb.com; gardner@neb.com; ludovic.sauguet@pasteur.fr

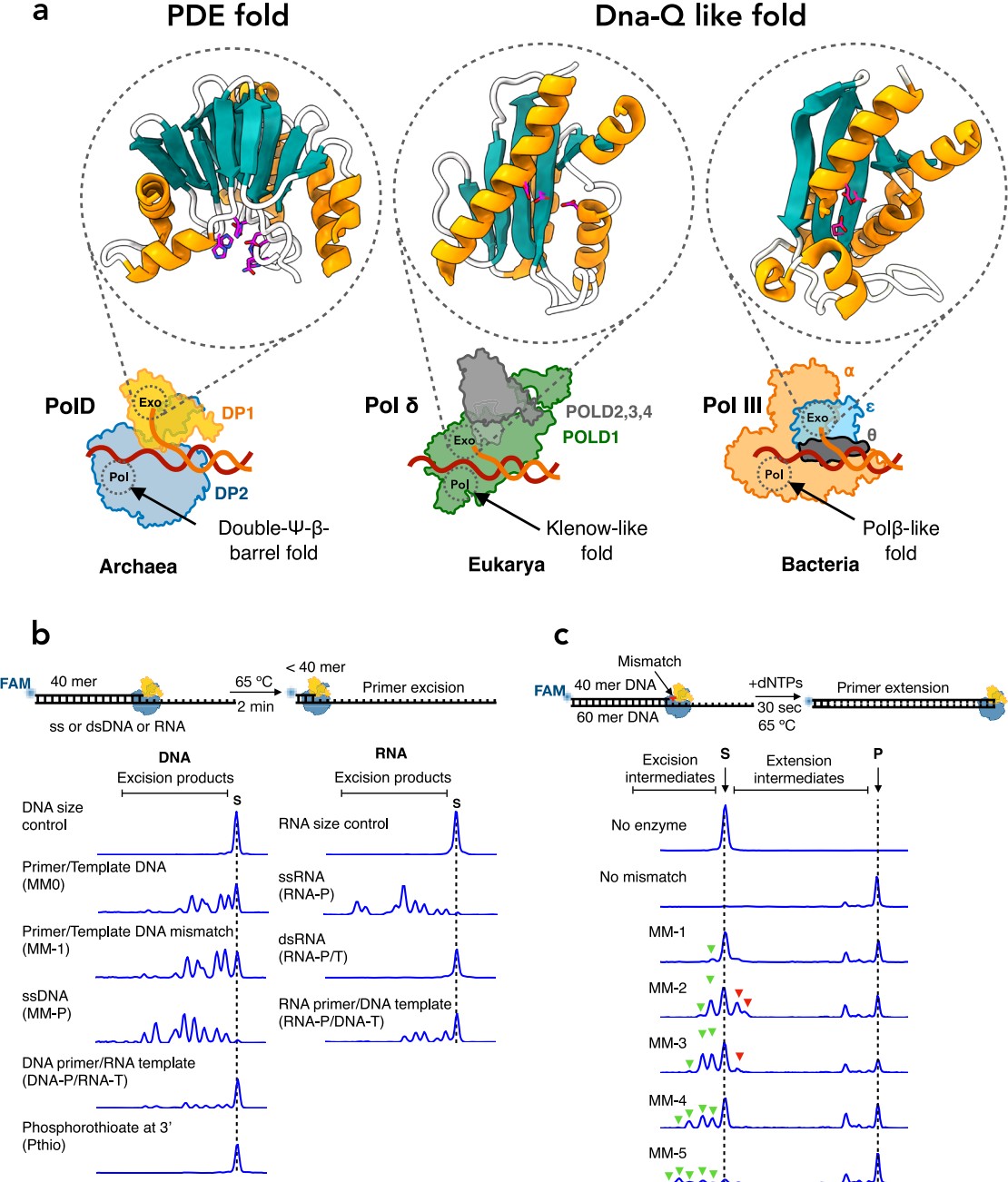

**Fig. 1 | Architecture and activity of the unique exonuclease domain of Family-D DNA polymerase. a** Schematic representation of replicative DNA polymerases from the three domains of life (Archaea-PolD (PDBid: 8PPT), Eukarya-Polδ (PDBid: 3IAY[79]), Bacteria-PolIII (PDBid: 5M1S[5])). Archaeal PolD exonuclease domain carries a unique PDE-fold, while Eukaryotic and most Bacterial exonuclease domains carry a Dna-Q-like fold. The exonuclease active site residues are shown in magenta. **b** Capillary electrophoresis (CE) excision assays. Top: Schematic of the primer excision assay. PolD was incubated with different single or double stranded substrates for 2 min at 65 °C. Bottom: CE traces Displaying excision activity of PolD on various DNA and RNA substrates. **c** CE primer extension assays. Top: Schematic of primer extension assays with substrates containing mismatches. PolD was incubated with different mismatch containing primer/template constructs for 30 s in the presence of dNTPs at 65 °C. Bottom: CE displaying excision intermediates (green arrowheads), extension intermediates (red arrowheads) and full-length product (the dotted line 'P'). Notably, extension pausing is observed at short timepoints when the polymerase reaches the end of the template. Source data are provided as a source data file.

subunit, contains a double-Ψ-β-barrel fold and carries the polymerase activity. Recent structural studies have shown that both polymerase and proofreading active sites differ from other structurally characterized DNA polymerases[25]. Before the first crystal structure of PolD was determined[25], the double-ψ-β-barrel catalytic core, which is housed within the DP2 subunit, was thought to be exclusively present in RNA polymerases[26]. DP1 possesses two main domains, an oligonucleotide binding (OB) domain, and the Mre11-like domain[27]. The latter

carries the PDE exonuclease catalytic core (Fig. 1a) that contains histidine, aspartate and asparagine residues that coordinate two metal ions required for exonuclease activity[25,27]. While PolD is the only DNAP that contains a PDE fold, this fold is also present in other enzymes that cleave phosphodiester bonds such as Mre11[28], LpxH[29] and calcineurin[30].

However, a detailed structural and functional understanding of how PolD utilizes the PDE catalytic core of DP1 for exonuclease

proofreading activity is currently lacking. In this work, we utilize cryo-EM and biochemical assays to obtain a comprehensive understanding of the proofreading mechanism of *Pyrococcus abyssi (P. abyssi)* PolD. For the first time, we present high resolution cryo-EM structures of PolD in the proofreading mode with single stranded DNA (ssDNA) stabilized within DP1. These structures reveal a variety of structural features responsible for correct orientation and stabilization of ssDNA during proofreading by this unique exonuclease active site. In addition to these structures, functional analysis of various PolD mutants provide insight into a previously unknown editing mechanism.

## Results

### Exonuclease and proofreading activities of PolD from *Pyrococcus abyssi*

Similar to other bacterial and eukaryotic repDNAPs, previous studies have shown that PolD possesses 3′–5′ exonuclease proofreading activities[24,31,32]. The DP1 exonuclease subunit from *Methanocaldococcus jannaschii* was previously shown to degrade ssDNA substrates and primer/template substrates containing mismatches[33]. Additionally, DP1 was shown to play a role in facilitating accurate DNA replication, as its inactivation lead to a reduction in replication fidelity[34].

To assess the exonuclease activity of PolD from *P. abyssi*, capillary electrophoresis (CE) based primer excision assays were performed using recombinantly expressed PolD and various single and double-stranded substrates (Fig. 1b). The highest excision activity of PolD was observed on single-stranded DNA and RNA substrates, as there was complete degradation of the primer substrates, demonstrating that PolD has both deoxyriboexonuclease and riboexonuclease activities. For the two double-stranded DNA primer/DNA template substrates, we observed less excision activity compared to ssDNA. Furthermore, the presence of a mismatch enhanced the exonuclease activity of PolD, as we observed more conversion of substrate to product for the mismatch-containing dsDNA compared to fully matched dsDNA. Minimal PolD exonuclease activity was observed when a DNA primer was annealed to an RNA template. However, higher exonuclease activity was observed when the template consisted of DNA and the primer was composed of RNA. Remarkably, no exonuclease activity was observed for a substrate with an RNA primer and RNA template. Furthermore, no excision products were detected when a DNA substrate containing a single terminal 3′ phosphorothioate was used, confirming that DP1 is an exclusive exonuclease.

To gain insight on the ability of PolD to sense a mismatch and engage exonuclease activity, a primer extension assay was performed in which PolD was incubated with dNTPs and various primer/template substrates containing a single mismatch at different primer positions. Fully paired primer/template DNA without a mismatch was used as a control. For this substrate, complete full-length extension products and no exonuclease products were observed, suggesting that PolD polymerase activity was favored over proofreading (Fig. 1c). Interestingly, when examining substrates with mismatches, both excision and extension intermediates were present. We observed a single excision intermediate on a substrate containing a terminal mismatch (MM-1), two excision intermediates on a substrate containing a mismatch at the −2 position, and three excision intermediates on a substrate containing a mismatch at the −3 position (Fig. 1c). This pattern continued for mismatches present at −4 (MM-4) and −5 (MM-5) positions, although excision intermediates were less abundant compared to those observed for mismatches at −1, −2, and −3 positions. These results suggest PolD senses a mismatch as far as 5 nucleotides back from the end of the primer and engages proofreading exonuclease activity to remove the mismatch. In addition to excision intermediates, we also observed the accumulation of one or two extension intermediates for MM-3 and MM-2 substrates, respectively (Fig. 1c), suggesting that a subpopulation of the mismatched primers was partially extended without proofreading occurring. We observed minor extension

intermediates for MM-1, MM-4 and MM-5 substrates, suggesting that PolD either properly removed the mismatch followed by extension to full length repaired product, or PolD readily bypassed the mismatch to create full length mismatch containing product. We speculate that PolD readily repairs mismatches at the −1 position (MM-1) but bypasses a certain percentage of mismatches at −4 and −5 positions.

### Cryo-EM structures of proofreading PolD

Single particle cryo-EM was used to understand the structural basis of proofreading by PolD. A catalytically inactive exonuclease variant of *P. abyssi* PolD was chosen (H451A^DP1) to prevent active digestion of DNA substrates (hereafter named PolD^exo-). Two primer/template (P/T) DNA duplexes containing mismatches were designed to trap the polymerase in the proofreading mode. The first P/T construct contained three consecutive thymine mismatches at the 3′-end of the primer, while the second substrate contained a single G:A mismatch at the −2 position of the primer. Proliferating Cell Nuclear Antigen (PCNA, the DNA sliding Clamp) was added to stabilize the PolD/DNA complex. For each substrate, cryo-EM structures of the complete PolD holoenzyme (DP1:DP2) along with PCNA were determined in the act of DNA proofreading with resolutions varying from 2.9 to 3 Å (Fig. 2a, b and Supplementary Figs. 1–3). The statistics of refinement and geometry of the structures are detailed in Supplementary Table 1. As expected, the PolD holoenzyme consists of two subunits: the DP2 polymerase subunit, and the DP1 exonuclease subunit. The two subunits are connected through a C-terminal helical bundle of DP1 and Zn-finger-III (Zn-III) motif found in the Clamp-1 domain of DP2. The holoenzyme is also linked to PCNA through its internal PIP-box (PCNA-Interacting peptide), present on the DP2 subunit[35]. Metaphorically, the heterodimeric 3D structure of PolD resembles the shape of a claw, where the Clamp-1 and Clamp-2 are the main domains that encircle the DNA duplex (Fig. 2a–c).

When PolD is bound to mismatched DNA, a heterogeneous spectrum of 3D conformations can be observed, where DP1 pivots around its interface with DP2 (Supplementary Figs. 1, 2, and 4). From this heterogenous population of particles, we isolated a closed conformation of the claw, referred to as the PolD-Exo complex in subsequent discussions, where the 3′ terminal nucleotide of the primer is within the exonuclease active site (Figs. 2c–f, 3b and Supplementary Fig. 5). Additionally, we observed multiple intermediate conformations of the open claw, characterized by increased flexibility in several DNA-binding domains, where the 3′ terminal nucleotide of the primer does not fully reach the exonuclease active site (Fig. 2g and Supplementary Fig. 4). The observation of the terminal nucleotide within the exonuclease active site in the closed exo conformation, and its absence in the open intermediate conformation, suggests that the transition from open to close conformation modulates PolD proofreading activity (Supplementary movie 1).

### Multiple interactions stabilize DNA in the PolD-Exo complex

The PolD-Exo complex structure reveals, with an unprecedented level of detail, the side chain interactions of the DP1 exonuclease and DP2 polymerase subunits of PolD with the P/T DNA duplex (Figs. 2d–f, and 3a). For the first time, we observed ssDNA in the DP1 domain. The conformation of bound DNA appears to be stabilized by three distinct sets of interactions. Firstly, the Clamp-1 domain of DP2 establishes strong interactions with the phosphate backbone located at the first minor groove of the DNA (Fig. 2d, e). These interactions distort the B-form helix of the DNA, suggesting it facilitates the melting of the primer from the template strand (Fig. 2h). Secondly, the N-terminal region of DP2 contains a K Homology (KH) domain (Fig. 2e)[36,37], that due to its location at the P/T fork junction, suggests it is involved in the separation of the two DNA strands and modulates DNA reannealing. Additionally, the KH domain is strategically

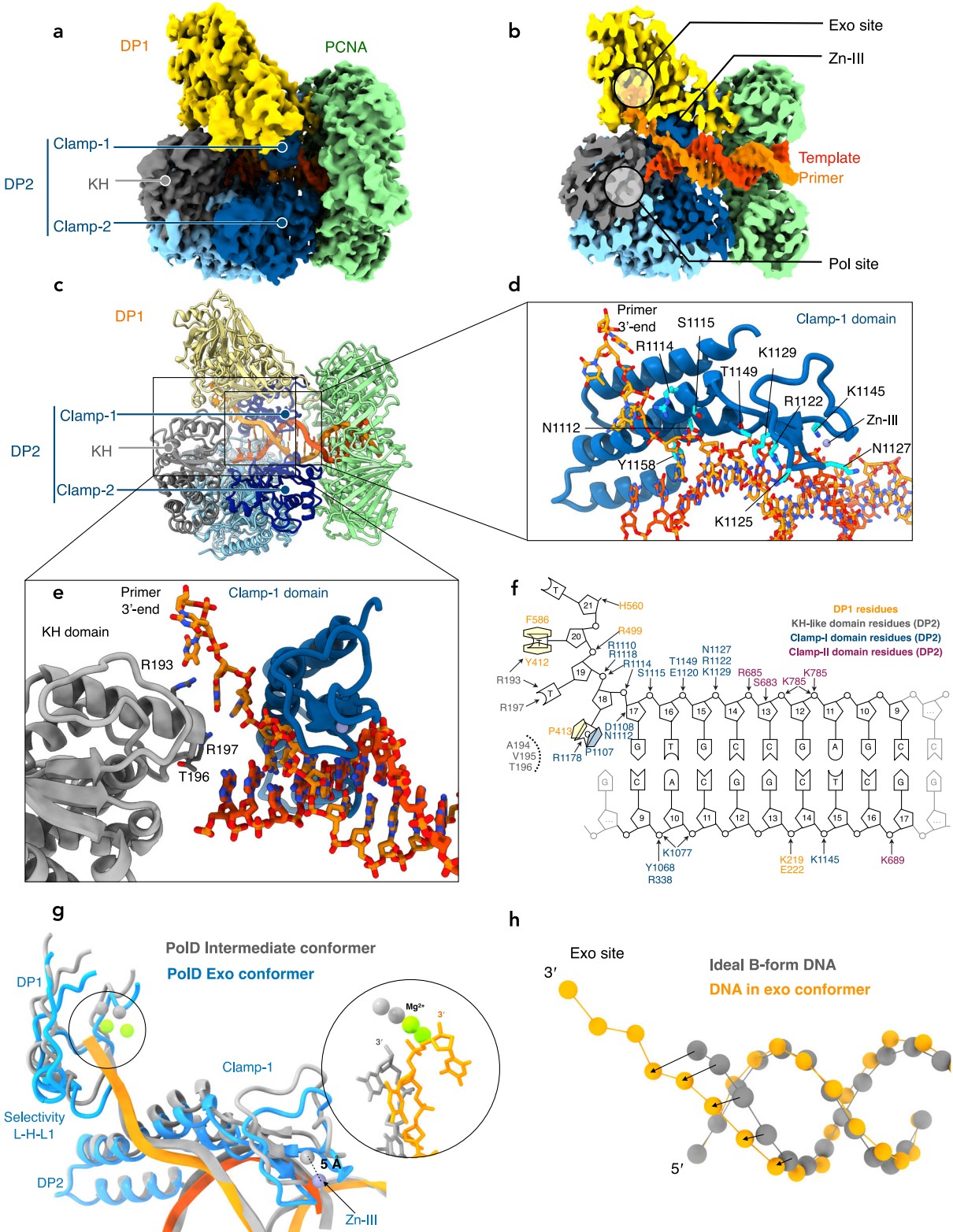

**Fig. 2 | Cryo-EM structure of proofreading PolD. a** Cryo-EM map of PolD-PCNA Exo mode with a DNA substrate containing three consecutive thymine mismatches. **b** Cutaway front view of the cryo-EM map highlighting the location of the DNA substrate with respect to the polymerase and exonuclease active sites. The primer and template of the DNA substrate are represented in orange and red, respectively. **c** Cartoon representation of the PolD-Exo complex showing the different DNA-binding domains **d** Focus on the contacts between the DNA duplex and the Clamp-1 domain, as well as the Zn-III domain of PolD. The side chains of DNA-contacting residues are represented in cyan. **e** Focus on the Clamp-I and KH domains that shuttle the primer strand toward the DP1 exonuclease domain. DNA-contacting residues from the KH domain are shown in gray. **f** Schematics of protein-DNA contact map of the PolD-exo structure. **g** Overlaid exo and intermediate conformers of PolD highlighting the structural changes in the Clamp-1 and Zn-III domains of DP2, as well as the DP1 exo-channel. These structural changes are also illustrated in Supplementary Movie 1. **h** Superimposition of an ideal B-form DNA helix (gray) to the forked DNA duplex in the PolD Exo mode model (orange).

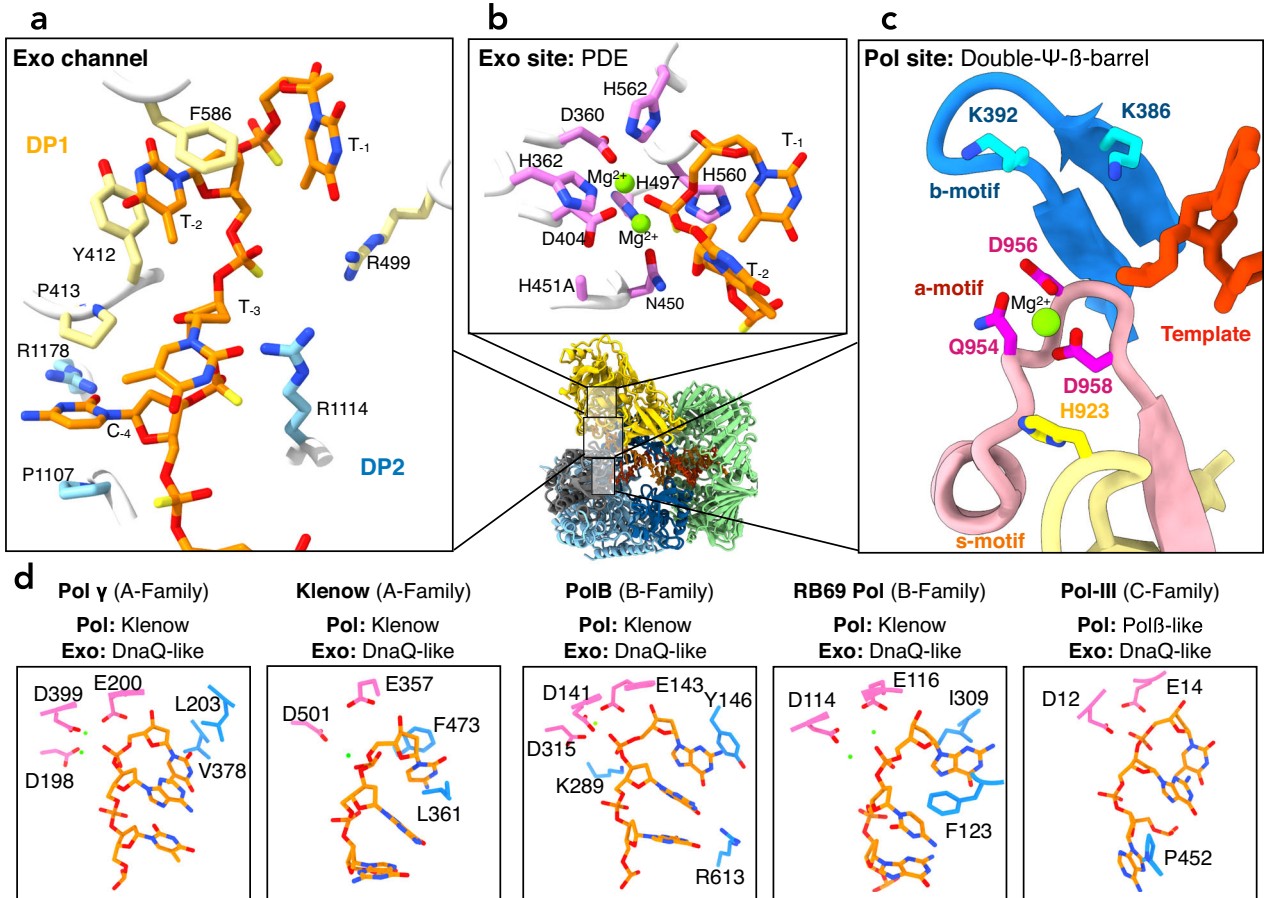

**Fig. 3 | Structure of the non-canonical exonuclease active site of PolD. a** Focus on the sidechains of the residues that line the PolD exo channel. Residues from DP1 and DP2 are represented in yellow and blue, respectively. The DNA primer is shown in orange. **b** Focus on the sidechains of the PDE active site residues (show in pink). **c** Focus on the sidechains of the DNA polymerase active site residues. These residues are clustered within three motifs a, b, and s (acidic in pink, basic in blue and selectivity in yellow, respectively). The catalytic $Mg^{2+}$ ions are shown in green. **d** Side-by-side comparison of the exonuclease active sites of Polγ (PDBid: 8D42[18]), *E. coli* DNA polymerase-I Klenow fragment (PDBid: 1KLN[14]), *P.abyssi* PolB (PDBid: 4FLT[16]), RB69 DNAP (PDBid: 1CLQ[41]), and *E.coli* Pol-III (PDBid: 5M1S[5]).

positioned to restrict access of the primer to the polymerase active site, instead guiding it into the DP1 domain. The third set of interactions result from amino acids in a channel that spans DP2 and DP1 (Figs. 2f and 3a). This channel, hereafter named the exo channel, is formed by side chains of numerous residues in the DP1 subunit (Fig. 3a), as well as the KH and Clamp-1 domains of the DP2 subunit.

While in this work we aimed to characterize the proofreading mode of the PolD DP1 subunit, the PolD-exo structure also revealed a well-defined polymerase active site, and the three previously characterized catalytic motifs[38], a-, b-, and s-, have been resolved with unprecedented detail (Fig. 3). For the first time, the coordination of a catalytic $Mg^{2+}$ ion by the side chains of the catalytic triad residues ($N954^{DP2}$, $D956^{DP2}$, and $D958^{DP2}$) is observed in the active site of PolD[39] (Supplementary Fig. 3). The position of this $Mg^{2+}$ ion corresponds to the position of $Mg^{2+}$-A ion in double-Ψ-β-barrel RNA polymerases[40]. $Mg^{2+}$-A has been shown to coordinate residues Q954, D956, and D958 in the polymerase active site, while $Mg^{2+}$-B, which is not observed in these structures, is supplied by the incoming nucleotide. While the template strand points toward the polymerase active site, the four terminal nucleotides of the melted primer are shuttled toward the DP1 nuclease active site, located 45 Å away from the pol site (Fig. 3a–c).

### An exo-channel guides the primer strand toward the nuclease active site

In the PolD-Exo complex, single-stranded DNA primer is guided towards the exonuclease active site of DP1 through the exo channel

formed by side chains of numerous residues in the DP1 subunit (Fig. 3a), as well as the KH and Clamp-1 domains of the DP2 subunit. Importantly, the four terminal bases of the melted primer are shuttled through the exo channel, with the terminal nucleotide present within the exonuclease active site (Fig. 3b). The first unpaired base of the primer (position −4) is positioned between two conserved proline residues, $P413^{DP1}$ and $P1107^{DP2}$ (Fig. 3a). The side chains of these residues form a pocket at the DP1-DP2 interface that accommodates the nitrogenous base moiety. In this pocket, the guanidinium group of $R1178^{DP2}$ is pointing towards the unpaired base, suggesting it forms hydrogen bonds in a coplanar arrangement. Adjacent to the exonuclease phosphodiesterase motif, the −1 nucleotide base is stabilized through hydrophobic interactions with the side chains of $Y415^{DP1}$, $V593^{DP1}$, $F589^{DP1}$ and polar interactions with the side chain of $E502^{DP1}$. The base at position −2 is situated between a pair of aromatic residues, $F586^{DP1}$ and $Y412^{DP1}$, resulting in a sandwich that creates pi-stacking interactions. The melted bases present in the exo channel are also stabilized through polar interactions with residues from the DP2 KH domain. Specifically, the melted bases at positions −2 and −3 interact with the guanidinium group and the main chain of residue $R193^{DP2}$. Lastly, the phosphate moieties of the nucleotides at positions −2, −3, and −4 engage in electrostatic interactions with the side chains of residues $R499^{DP1}$ and $R1114^{DP2}$. These interactions contribute to the overall stability of the melted bases within the PolD-Exo complex.

It is worth noting, that in other DNA polymerase families (A, B, and C) that contain a DnaQ-like exonuclease fold, three bases of the DNA

primer are separated from the template and shuttled to the exonuclease active site[5,14,16,18,41]. The PolD-Exo complex demonstrates a distinctive separation of four base pairs of the DNA primer, and further highlights the unique features of the PolD-Exo complex in comparison to other DNA polymerase families (Fig. 3d). Additionally, in DnaQ-containing repDNAPs, the intramolecular pi-stacking interactions present in duplex DNA are mostly retained in the three bases described above[5,14,16–18,41], while these interactions are lost amongst the four bases present in the PolD exo channel. Instead, each of the four bases interacts independently with amino acid side chains in the exo channel (Fig. 3a). Thus, the binding mode of ssDNA in the exonuclease domain contrasts between PolD and other families of repDNAPs. The significance of these interactions is underscored by the considerable surface area involved in the interaction between PolD and the melted primer strand. In the PolD-Exo complex, this surface interaction measures approximately 845 Å². Comparatively, this value is notably higher than the surface interaction observed in the DnaQ-containing Polγ (426 Å²), RB69 (561 Å²) and Pol-III (540 Å²) DNA polymerases.

## The non-canonical exonuclease active site of PolD

The exonuclease active site of PolD exhibits a PDE-like fold, which distinguishes it from the DnaQ-like fold that is conventionally found in the exonuclease domain of other families of repDNAPs[13,19] (Figs. 1a and 3b). As expected, but now structurally observed, the active site of DP1 contains two metal ions coordinated by four histidine residues (H362$^{DP1}$, H497$^{DP1}$, H560$^{DP1}$, and H562$^{DP1}$), two aspartate residues (D360$^{DP1}$ and D404$^{DP1}$), and one asparagine residue (N450$^{DP1}$)[25]. As described previously, H451$^{DP1}$, a catalytically essential exonuclease residue, was mutated to alanine, to trap the polymerase in editing mode. Two $Mg^{2+}$ ions were modeled in the exonuclease site of PolD-Exo, as the cryo-EM structure was determined in the presence of $MgCl_2$ (Supplementary Fig. 3). Furthermore, the exonuclease activity of PolD from *P. abyssi* is stimulated in the presence of $Mg^{2+}$ ions (Supplementary Fig. 6), consistent with previous studies on PolD from different species[12,33,42]. The terminal nucleotide of the ssDNA substrate, which is located within the exonuclease active site, is present as a pre-catalytic complex, with the scissile bond positioned between the two $Mg^{2+}$ ions. This conformation enables hydrolysis in which the metal ions stabilize the phosphodiester bond for proper nucleophilic attack by an activated water molecule[30]. The ribose moiety of the nucleotide to be cleaved is stabilized through a hydrogen bond between the 3'OH group of the sugar and the carbonyl of the main chain of residue H560$^{DP1}$. The structure of the complex also reveals that there is no steric hindrance between the exonuclease active site and the hydroxyl group at the 2' position of the ribose, which is consistent with our digestion tests that revealed PolD can digest single-stranded DNA or RNA (Fig. 1b).

## Intermediate open-claw conformations of PolD

In addition to the PolD-Exo conformation, PolD adopts multiple intermediate states when bound to mismatched DNA that are characterized by a more open conformation of the claw (Supplementary movie 1). With the opening of the claw, the module formed by DP1, the Clamp-1 domain and Zn-III motif of DP2, moves away from the Clamp-2 domain, relaxing contacts with the DNA minor groove (Fig. 2g). Indeed, while the measured surface interaction between the primer strand and the polymerase is 1543 Å² for the PolD-exo conformer, it is only 1083 Å² for the intermediate conformer. These observations suggest that PolD-Exo promotes tight binding of the DNA, guiding the primer strand for digestion in the exonuclease active site. Intermediate open conformations are characterized by more relaxed DNA binding that allows PolD to scan DNA pairing. If the mismatch has been corrected, the primer can reanneal to the template and the polymerase returns to an extension mode. The exchange between these different conformations could explain the non-processive nature of the

exonuclease activity of PolD observed during elongation experiments with substrates containing mismatches at different positions (Fig. 1c).

## Pre steady-state single turnover kinetics of PolD and exo-channel mutants

The cryo-EM structures presented above revealed a set of residues that stabilize the DNA substrate in the exo channel of DP1 (Fig. 4a). To quantitatively assess the contribution of each residue in the exo channel to exonuclease activity, CE-based pre-steady-state single turnover kinetic assays were performed with high enzyme-to-substrate ratio. The substrate used for these assays was a 5' FAM-labeled DNA primer containing a phosphorothioate (Kin-P) at the −2 phosphodiester bond, limiting digestion to a single nucleotide and preventing multiple excision events (Fig. 4b, c). With this substrate design, the rate of nucleotide cleavage by PolD and exo channel mutants ($k_{obs}^{exo}$) can be measured. To understand the effects of primer/template substrates, the phosphorothioate-containing primer was also annealed to two different templates, resulting in the creation of a fully paired primer/template and a primer/template with a single terminal mismatch at the 3' end of the primer. To determine the $k_{obs}^{exo}$ values, the concentration of product was plotted as a function of time for each mutant and substrate (Fig. 4d, e). Note that mutants P413A$^{DP1}$, P1107A$^{DP2}$, and R1114A$^{DP2}$ were not included in this study as their excision profiles do not show a strong effect on the exonuclease activity (Supplementary Fig. 7) and mismatch bypass (See below).

The fastest rate of exonuclease activity for ssDNA substrate was observed for PolD$^{wt}$, with a $k_{obs}^{exo}$ of $8.1 \times 10^{-2}$ s$^{-1}$. PolD$^{wt}$ showed slower exonuclease activity on a paired P/T substrate ($k_{obs}^{exo}$ of $2.3 \times 10^{-2}$ s$^{-1}$). The rate of PolD$^{wt}$ exonuclease activity for a P/T substrate containing a mismatch is 2.3 times faster compared to a fully paired substrate and 1.5 times slower than the excision rate of a single stranded DNA substrate, demonstrating that a mismatch triggers the exonuclease activity of PolD and melting of the P/T is the rate limiting step of the proofreading activity[43]. Varying rates of exonuclease cleavage were observed for the exo channel mutants. The most significant reduction in exonuclease activity was observed when both residues of the exo channel, Y412$^{DP1}$ and F586$^{DP1}$ and the adjacent arginine, R499$^{DP1}$, were mutated to alanine, resulting in the complete loss of exonuclease activity on all substrates. Mutation of either aromatic residues of the exo channel, Y412$^{DP1}$ or F586$^{DP1}$, to alanine reduced ssDNA cleavage rate by half, and P/T cleavage rate by ~80%. Finally, for the R1178A$^{DP2}$ mutant, we observe a similar rate of ssDNA cleavage compared to wild-type, yet a ~60% reduction in cleavage of P/T substrates compared to wild-type. It is important to note that the above mutants retain an active PDE motif, and therefore the reduction of the exonuclease cleavage rate is likely due to improper binding of the substrate in the exo channel of the DP1 domain of PolD.

## Mismatch bypass of exo channel PolD mutants

PolD exonuclease kinetic experiments showed that the exo channel is lined with key amino acids that stabilize the ssDNA primer for proper exonuclease digestion. Next, we aimed to further understand the role of these amino acids on PolD mismatch bypass. A CE-based assay was designed, where PolD$^{wt}$, or PolD mutants, were incubated with dNTPs and a P/T substrate containing a single mismatch. If the mismatch was repaired by the proofreading activity of PolD, a BstBI restriction site was generated (Supplementary Table 2). If PolD bypassed the mismatch, no restriction site was generated. Following PolD extension, DNAs were then digested with BstBI to determine the fraction of bypassed versus repaired products (Fig. 5a–c).

As anticipated, for PolD$^{exo-}$, which lacks exonuclease activity, we observed 100% mismatch bypass, while PolD$^{wt}$, which retains exonuclease activity, bypassed 38% of mismatched substrates. All exo channel mutants, except for R1178A, showed higher mismatch bypass frequency compared to wild type. Mutants close to the P/T junction and far from the exonuclease active site, such as P413A$^{DP1}$, P1107A$^{DP2}$,

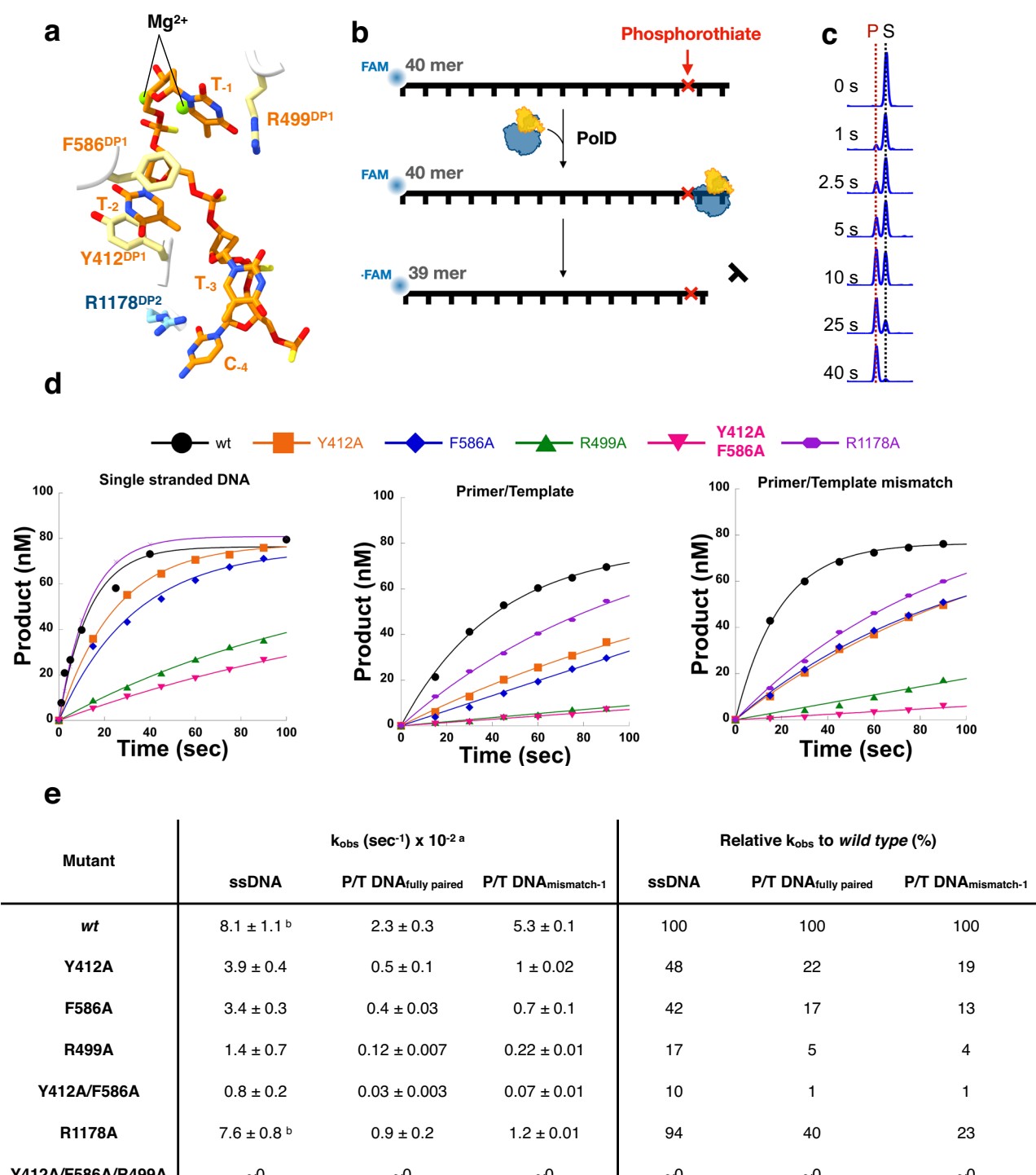

**Fig. 4 | Pre-steady-state exonuclease kinetics of PolD and exo-channel mutants.**
**a** Schematic of important exo channel residues shown in yellow (DP1) and blue (DP2). The melted primer strand is shown in orange. **b** Schematic of the pre-steady-state exonuclease reaction, where an excess of PolD (or PolD mutant) was rapidly mixed with substrate for different timepoints to allow for nucleotide excision. The DNA substrate contains a phosphorothioate at the −2 phosphodiester bond to ensure single nucleotide excision. **c** Representative CE displaying the conversion of 40mer substrate (S) to 39mer product (P) by single-nucleotide excision of PolD.

**d** Representative time courses for the three substrates tested: single stranded DNA (Kin-P), Primer/Template DNA (Kin-FM) and Primer/Template DNA containing a terminal mismatch (Kin-MM-1). Each trace corresponds to a mutant that is indicated by the legend on top of the panel. **e** Exonuclease rates values determined for PolD wild-type and the exo-channel mutants, for the different DNA substrates. Footnotes: a: The displayed rates are the average of three experimental replicates ($n = 3$) ± standard deviation. b: Experiments performed utilizing a rapid chemical quench instrument. Source data are provided as a source data file.

and R1114A[DP2], bypassed ~ 50% of mismatched substrates, while residues close to the active site and far from the P/T junction had a more severe effect. For example, mutants Y412A[DP1] and F586A[DP1] bypassed ~70% and 83% of mismatched substrates, respectively, while mutation of R499A[DP1] was the single site mutant that showed the largest effect on PolD mismatch bypass, resulting in 85% mismatch bypass. Double mutant Y412A[DP1]/F586A[DP1] bypassed 96% of mismatches, while the triple mutant Y412A[DP1]/F586A[DP1]/R499A[DP1] showed the same phenotype

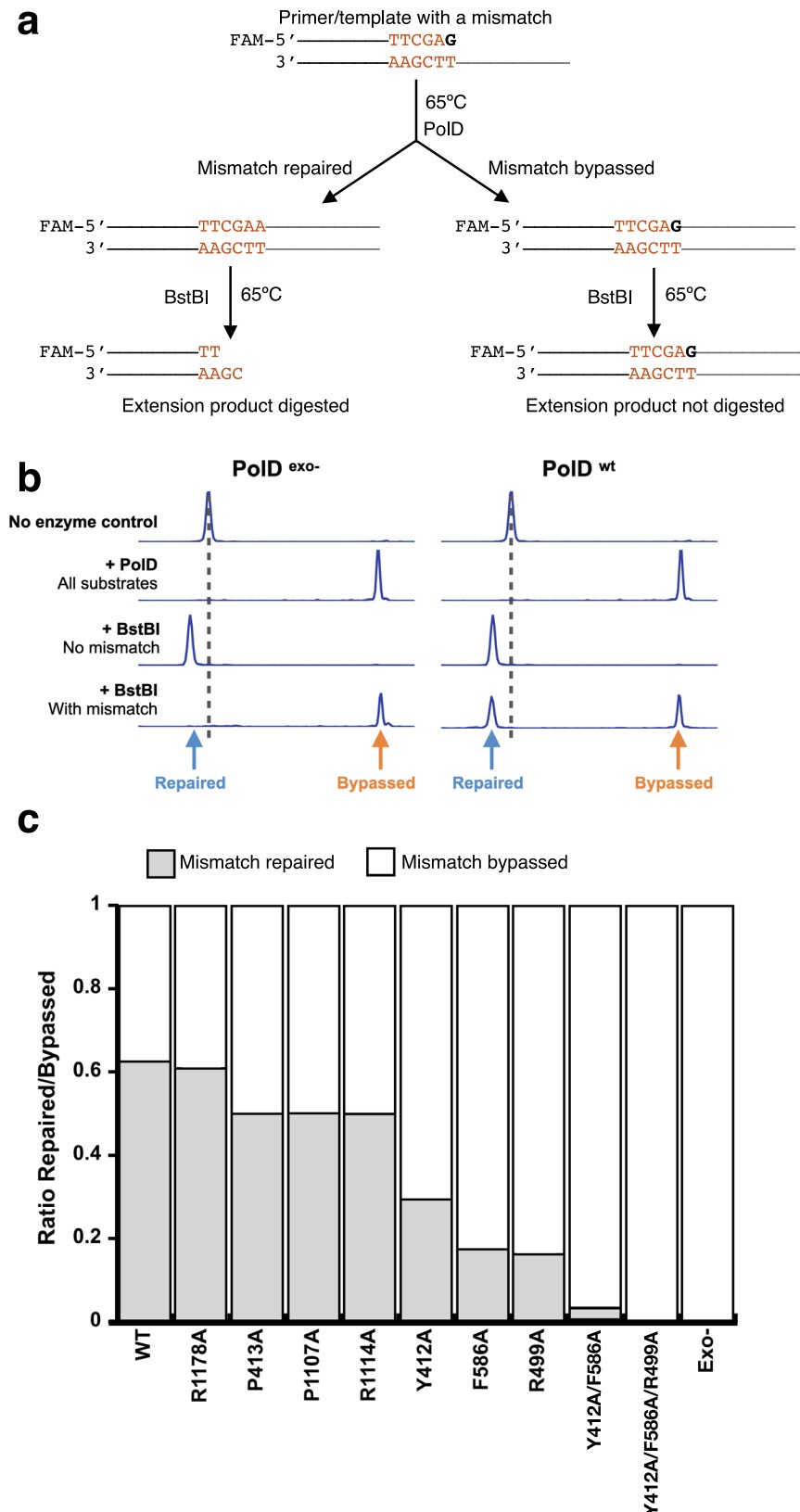

as PolD[exo-] and bypassed 100% of mismatches. In agreement with exonuclease kinetic reactions, we observed the highest amount of mismatch bypass with PolD mutants that showed the slowest exonuclease cleavage rate, and the least amount of bypass with PolDs that had the highest exonuclease cleavage rate. Therefore, the exonuclease cleavage rate of PolD (or PolD mutants) dictates mismatch bypass.

## Discussion

All repDNAPs have evolved a proofreading exonuclease activity carried out by a distinct active site separate from the polymerase active site[1]. The structures of several representative DNA polymerases have been determined while engaged in proofreading, leading to a better understanding of the molecular mechanisms involved in this

**Fig. 5 | Mismatch bypass by PolD and exo-channel mutants. a** Schematic of the mismatch bypass assay. A primer/template substrate containing a single mismatch is fully extended by PolD or exo-channel mutants. Extended products were digested with BstBI and fragments were analyzed by CE to determine mismatch bypass ratios. **b** Representative CE traces of the mismatch bypass reaction using PolD[exo-] and PolD[wt]. No enzyme control traces correspond to the substrate before extension, +PolD traces correspond to fully extended products. The +BstBI No Mismatch traces correspond to digestion products of a substrate that does not contain a mismatch, therefore has an intact BstBI restriction site leading to full digestion and the Repaired product peak. The +BstBI With Mismatch traces correspond to digestion products of extended primers with a mismatch. For PolD[exo-], the extended products retain the mismatch resulting in no digestion by BstBI. For PolD[wt], two different peaks show two different populations of digestion products, repaired or bypassed. **c** Ratio of repaired/bypassed products for PolD[wt] and exo-channel mutants. The experiment was replicated two times independently and provided the same results. Source data are provided as a source data file.

fundamental process. Hence, the structures of several DnaQ-like containing repDNAPs have been determined in the exonuclease mode, including the Klenow fragment of Pol-I (Family A)[14], Polγ (Family A)[18], RB69 (Family B)[41], archaeal PolB (Family B)[16], and Pol-III (Family C)[5,17].

## The proofreading active site of PolD is unique among DNA polymerases

Our group previously reported that the PolD double-Ψ-β-barrel polymerase catalytic core differs from the Klenow-like and Polβ-like folds found in all other families of repDNAPs[25]. Further, while other repDNAPs share a DnaQ-like active exonuclease site, PolD carries out proofreading utilizing an Mre11-like fold. In this study, we have determined the first structure of PolD captured in a proofreading mode and extensively characterized its exonuclease activity, further highlighting that PolD is structurally distinct from other repDNAPs.

First, the PDE editing domain of PolD possesses a completely different fold compared to the canonical DnaQ fold (Fig. 1a). The nature of the residues that coordinate the catalytic divalent ions (four histidines, two aspartates and one asparagine in the PDE active site) differs from the two or three catalytic acidic residues found in DnaQ-like nucleases. Second, while D-family DNA polymerases have evolved a narrow exo-channel that binds four bases of the melted primer strand, all other repDNAPs show a mechanism in which three bases of the DNA primer are separated from the template and shuttled to the exonuclease active site. Furthermore, the PolD-Exo complex demonstrates a distinctive binding mode of the melted nucleotides. Indeed, while in DnaQ-containing repDNAPs, the intramolecular pi-stacking interactions present in duplex DNA are at least partly retained in the three melted bases[5,14,16-18,41], these interactions are lost amongst the four bases present in the PolD exo channel, and compensated by extensive interactions with amino acid side chains in the exo channel (Fig. 3). Importantly, our functional data illustrate the importance of this exo channel. Mutations on these non-catalytic residues greatly affect the exonuclease activity and the mismatch bypass, demonstrating that correct binding of the 3′ DNA end is as important as the catalytic activity of the PDE active site. Third, the cryo-EM structure of the PolD-exo conformer also reveals that the primer strand is melted and guided toward the exonuclease active site through a mechanism which involves a KH domain and a Zn-finger domain that are structurally unrelated to the finger and thumb domains described in other families of repDNAPs. For example, a beta-hairpin originating from the exonuclease domain of most family -B DNA polymerases has been shown to play a key role in the melting of the primer and template strands[44,45]. However, this beta-hairpin structure is not present in PolD, which highlights the unique structural features of PolD. Instead, the KH domain uniquely present in PolD, is located at the DNA fork and acts as a gate preventing both the reannealing of the primer to the template and access to the polymerase active site.

## A unified mechanism for proofreading

This study indicates that, despite its distinctive three-dimensional structure, the proofreading mechanism of PolD shares multiple features with other DNA polymerase families. First, as in all other families of repDNAPs, the polymerization and editing enzymatic activities are performed by two separated active sites. In fact, this feature was particularly relevant for PolD, which shares the same catalytic core as RNA polymerases, where it has been shown that both synthesis and editing of the nascent mRNA occur at the same double-Ψ-β-barrel catalytic core[46-48]. Although PolD also possesses a similar double-Ψ-β-barrel catalytic core, our study unambiguously demonstrates that the proofreading activity of PolD is exclusively performed at a separate active site. The polymerase and exonuclease active sites are 45 Å apart in PolD, a distance similar to that observed in other repDNAPs (Pol-I (35 Å), Polγ (38 Å), PolB (26 Å), RB69 (37 Å), and Pol-III (60 Å)), where it has been demonstrated that the switching between polymerization and proofreading modes involves both DNA movement and structural rearrangement of the polymerase. Although PolD has evolved different domains than those found in other families of DNA polymerases, the PolD-exo structure reveals a conserved mechanism in which the 3′ end of the primer is separated from the template strand and bent towards the exonuclease site.

Like the structural diversity, the rates of the exonuclease activities of repDNAPs also varies among the different families of polymerases[43,49-54]. Although the reported kinetic rate for *P.abyssi* PolD ($8.1 \times 10^{-2}$ s$^{-1}$) is slower compared to *Thermococcus 9°N* PolD (2.5 s$^{-1}$)[55], likely due to differences in experimental conditions and protein sequence, it is similar to those reported for other cellular repDNAPs, such as Pol-ε ($1.5 \times 10^{-2}$ s$^{-1}$)[50] and Pol-I ($12 \times 10^{-2}$ s$^{-1}$)[56]. Yet, these values are slower than the ones reported for viral monomeric DNAPs, such as T7 and T4 DNAPs (105 and 200 s$^{-1}$, respectively)[49,52]. Notably, these assays are carried out at different conditions (pH, temperature and reaction buffers), which have been shown to strongly influence the observable exonuclease performance of DNAPs[53]. Additionally, we observe a trend common to all polymerases where the maximum rate of the exonuclease activity is observed for ssDNA substrates, followed by substrates with mismatches and fully paired substrates[50-52,55]. Overall, like other polymerases, the rate limiting steps for exonuclease cleavage by PolD is the recognition the mismatch at the polymerase active site, melting, and shuttling of the primer end to the exo site (Fig. 4d, e). Proofreading by repDNAPs has been suggested to follow a sequential pattern that starts with mismatch detection, followed by switching to the exonuclease active site, exonuclease reaction, and ends with switching back to the polymerase mode to check for correct geometry. Such sequential mechanism has been reported for several DNAPs. The exonuclease active site of T4 DNAP digests one terminal nucleotide at a time; after each excision event, the primer is reannealed at the polymerase active site and examined for correct geometry to determine if another excision cycle is necessary[49]. A similar switching mechanism between polymerization and exonuclease modes has also been observed for Polε[45]. Additionally, RB69 DNA polymerase crystallized with DNA primer with a template containing an abasic site, shows four different conformations in both exo and pol modes[57]. At a structural level, this mechanism implies that DNA polymerases can exist, not only in exo and pol states, but also in multiple intermediate states. Recent structural studies on Polγ revealed this enzyme exists in polymerase, exonuclease, as well as in an intermediate conformers[58]. Additionally, Polγ utilizes a dual checkpoint mechanism to sense nucleotide misincorporation and initiate proofreading[18]. Similarly, we have shown that PolD can exist in both editing and intermediate conformers. These intermediate conformers are characterized by a more relaxed DNA binding that may allow PolD to

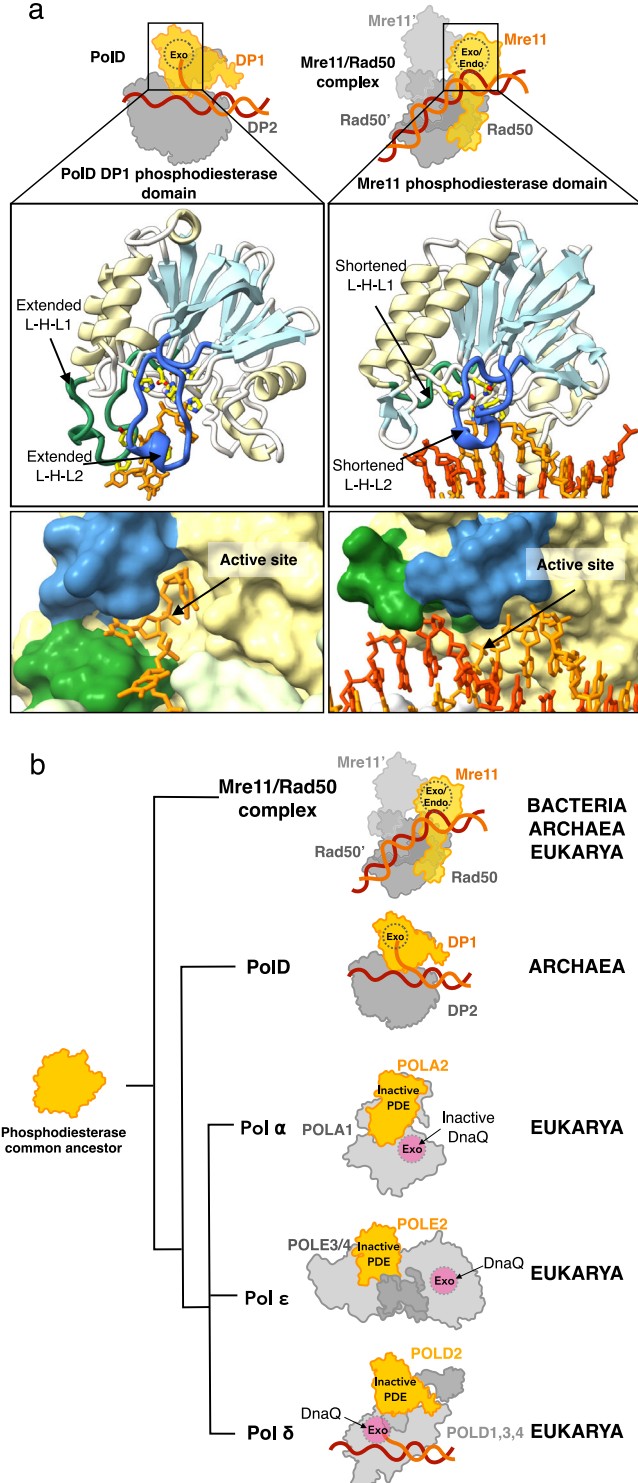

**Fig. 6 | Evolutionary implications of the PDE-fold across proteins and Domains of life. a** Top: Schematic representations showing the nuclease active sites in PolD (left) and Mre11/Rad50 (right). Middle: Cartoon representation of the PDE domain of PolD (left) and Mre11/Rad50 (right). The α-helices are represented in yellow and the β-sheets in light blue to illustrate the shared structural topology of their PDE domains. The loop-helix-loop motifs 1 and 2 (L-H-L 1 and 2) are highlighted in green and dark blue, respectively. Bottom: Surface representation of the PolD exo channel (left) and the corresponding region in Mre11/Rad50 (right). The ssDNA substrate is represented in sticks (orange). The active site location is indicated by the black arrow and the L-H-L 1 and 2 motifs are highlighted in green and dark blue, respectively. **b** Putative evolutionary scenario of the PDE domain in Mre11 and DNA polymerases from a common ancestor. An active PDE motif has been retained in archaeal PolD, as well as in Mre11, a ubiquitous exo/endonuclease found in all three domains of life. Through evolution, the PDE motif became inactivated in eukaryotic repDNAPs as they acquired the Dna-Q-like exonuclease domain.

While the involvement of an active PDE-like domain in a DNA polymerase and its participation in proofreading activity is specific to family D DNA polymerases, the PDE fold is found in numerous nucleases, particularly in Mre11, a key player in genome maintenance. Mre11 is an exo/endonuclease that exists in all three domains of life, where it is associated with Rad50 and participates in double-strand break repair pathways[28,60]. Structures of the Mre11/Rad50 complex, captured during the digestion of different DNA substrates, have recently been determined[61], allowing for a detailed comparative study of the molecular basis of DNA binding specificity in PolD and Mre11. While Mre11 has dual endonuclease and exonuclease activities[60], it has been shown that the DP1 subunit of PolD exhibits strict exonuclease activity (Fig. 1b). The PDE domains of *P. abyssi* DP1 and *E.coli* Mre11-show a strong overlap (Fig. 6a). The residues involved in coordinating metal ions in both enzymes are remarkably conserved, and both nucleases position the phosphate group ready for cleavage in an identical manner, indicating a shared hydrolysis mechanism. However, comparing the active sites of PolD and Mre11 nucleases reveals how PolD has evolved a unique DNA binding site that allows it to discriminate between ssDNA and dsDNA. Indeed, the structure shows how PolD prevents the access of dsDNA substrates to the DP1 exonuclease active site: two exonuclease selectivity loops located at the heart of the PDE domain restrict access to the active site to ssDNA. These two extended loops are inserted between residues 403[DP1] and 423[DP1], and 577[DP1] and 597[DP1] and contain residues that specifically interact with the ssDNA primer. Each one specifically carries one of the two aromatic residues, Y415[DP1] and F589[DP1], which sandwich the penultimate base of the primer and contribute to the formation of the exo channel. In addition, the structure provides a clear demonstration of how the spatial arrangement of these two motifs would prevent the binding of DNA duplex in the nuclease active site. In the structure of Mre11 these two loops are shortened, allowing the binding of dsDNA substrates (Fig. 6a).

Altogether, this work highlights the complex evolution of the PDE nuclease fold, which is broadly distributed across the three domains of life. The Mre11 and PolD PDE domains most likely evolved from a common ancestor, which may have existed in the Last Universal Common Ancestor (LUCA). This ancestral nuclease has acquired different substrate specificities, exo- or endonuclease, and have been recruited by different DNA binding proteins, to achieve diverse biological functions in genome maintenance, such as DNA recombination (Mre11/Rad50) or DNA proofreading (PolD) (Fig. 6b). While Mre11/Rad50 is a universal genome guardian in all three domains of life, the polymerase-associated PDE proofreading domain is only found in Archaea in its active form. Indeed, while the architecture of the PDE domain of PolD has been conserved in the B-subunit of all three eukaryotic repDNAPs[10,27] (Polα, Polδ, and Polε), its nuclease activity has been lost over the course of evolution. In eukaryotes, the ancestral polymerase-associated PDE has thus been replaced by a DnaQ nuclease

scan DNA pairing. Interestingly, in Polγ, the transition from replication to error editing has also been shown to be accompanied by increased dynamics in both DNA and the polymerase, in which the enzyme backtracks to shuttle the mismatch-containing primer terminus to the exo site for editing[18]. Finally, our observations of primer extension with substrates containing buried mismatches (Fig. 1b), are consistent with kinetic studies carried out with T7 DNAP, where mismatches at positions −1 to −3 are repaired more efficiently than buried mismatched at positions −4 to −10[59]. Altogether, these observations show a convergent evolution phenomenon across DNA polymerases that carry out proofreading.

fold in all repDNAPs. Importantly DP1 and the B-subunit of eukaryal repDNA polymerases share a role in recruiting different replication factors such as GINS[62,63], and the Primase[64]. Similarly, in most bacterial species, PHP, another ancestral nuclease domain, has also been shown to be inactivated and replaced by a DnaQ nuclease fold[65]. This study sheds light on the convergent evolution of structurally diverse families of DNA polymerases, and the domain acquisition and exchange that occurred during the evolution of the replisome in the three domains of life.

## Methods

### DNA substrates and mutants

All DNA substrates used in this work were synthesized by Eurogentec (Seraing, Belgium) or Integrated DNA technologies (Coralville, IA) and the sequences are detailed in Supplementary Table 2. PolD mutants created in this work are shown in Supplementary Table 3.

### Protein expression and purification for cryo-EM

The clone (pLB047) containing DP1^exo- (H451A) and DP2 subunits of PolD, from *P.abyssi* was previously described, and optimized for expression in *E.coli*[25]. For cryo-EM, *E.coli* T7 express competent cells (C2566 New England Biolabs Inc., Ipswich, MA) were transformed with pLB047, as indicated by the manufacturer. Transformed cells were plated on an LB-Kanamycin (LBK) agar plate and grown overnight at 37 °C. A 200 mL pre-culture, grown overnight at 37 °C in LBK, was used to inoculate 8 L of LBK. The inoculated cultures were grown at 37 °C until an $OD_{600}$ of 0.6. Expression of PolD^Exo- was induced with 1 mM final IPTG (0008-B Euromedex) and growth was continued overnight at 20 °C. Induced cells were harvested by centrifugation at $3500 \times g$ for 30 min and resuspended with lysis buffer (25 mM HEPES (H7006 Sigma), 300 mM NaCl (S9888 Sigma), 2.5 µM MgCl$_2$ (M8266 Sigma), 2.5 µM Zn(CH$_3$CO$_2$)$_2$) (Z0625 Sigma) supplemented with 1x Pierce™ Protease Inhibitor tablets, EDTA-free (15677308 Thermo scientific) and 2 µL Benzonase Nuclease (E1014 Millipore). Resuspended cells were lysed with a Cell-disruptor (Constant systems LTD, Northants, UK). Lysate was incubated at 60 °C for 10 min followed by pelleting at $20,000 \times g$ for 20 min. Lysate supernatant was filtered with a 0.33 µm filter unit and loaded onto a 5 mL Ni-NTA column (17528601 Cytiva) pre-equilibrated with buffer A (20 mM Na-HEPES pH 8, 300 mM NaCl, 20 mM Imidazole (I202 Sigma) pH8), and eluted with buffer B (20 mM Na-HEPES pH8, 300 mM NaCl, 500 mM imidazole) with an imidazole gradient from 20 mM to 500 mM imidazole. Fractions were run on a 4–20% Tris-Glycine gel (1610732 BioRad) to confirm the presence and purity of PolD DP1^Exo- and DP2. Fractions containing the protein were pooled and diluted 5x in buffer C (20 mM HEPES pH8). The fractions were loaded onto a heparin column (17040703 Cytiva) pre-equilibrated with buffer C and eluted with buffer D (20 mM Na-HEPES, 1 M NaCl) with a NaCl gradient from 50 to 1000 mM. Fractions were run on a 4–20% Tris-Glycine gel to confirm the presence and purity of PolD DP1^Exo- and DP2. Fractions containing the protein were concentrated with a 100 kDa MWCO filter unit (UFC910024 Amicon). Concentrated fractions were loaded onto a Superose 6 column (29091596 Cytiva) in buffer E (20 mM HEPES pH 8, 150 mM NaCl, 2 mM MgCl$_2$). Fractions were run on a 4–20% Tris-Glycine gel to confirm the presence and purity of PolD DP1^Exo- and DP2. The fraction containing the highest concentration of PolD was used for cryo-EM sample preparation (Supplementary Fig. 8). PCNA was cloned, expressed and purified as previously described[35].

### Site directed mutagenesis, expression and purification of PolD for biochemical characterization

All PolD mutants (Supplementary Table 3) were generated using pLB047 and a Q5 Site-Directed mutagenesis kit (E0554 New England Biolabs Inc., Ipswich MA) as described by the manufacturer. Primers were designed using the NEBaseChanger tool (http://nebasechanger.neb.com).

All mutants were confirmed by Sanger sequencing. For expression of PolD^wt, PolD^Exo-, and other mutants, T7 express or NiCo21 competent cells (C2529 New England Biolabs Inc., Ipswich, MA) were transformed with each clone, as indicated by the manufacturer. For each mutant, transformed cells were plated on an LBK agar plate and grown overnight at 37 °C. A 50 mL pre-culture, grown overnight at 37 °C in LBK, was used to inoculate 1 L of LBK. The inoculated cultures were grown at 37 °C until an $OD_{600}$ of 0.5. Expression of each PolD was induced with 0.4 mM final IPTG and growth was continued for 3 h at 37 °C. Induced cells were harvested by centrifugation at $4000 \times g$ for 30 min and resuspended with buffer A, followed by lysis using a constant cell disruptor (constant systems LTD, Northants, UK). Lysates were incubated at 80 °C for 20 min followed by centrifugation at $20,000 \times g$ for 30 min. For each mutant, the supernatant was loaded onto a DEAE 16/10 FF column (28936541 Cytiva) and flowthrough was collected and loaded onto a 1 mL HisTrap HP column (17524701 Cytiva) pre-equilibrated with buffer A and eluted with buffer B (20 mM Na-HEPES pH8, 300 mM NaCl, 500 mM imidazole) with an imidazole gradient from 20 mM to 500 mM. Fractions were run on a 4–20% Tris-Glycine gel to confirm the presence and purity of each mutant of PolD. Fractions containing the protein were pooled and dialyzed into storage buffer (100 mM KCl, 10 mM TrisHCl (pH 7.4) 1 mM DTT, 0.1 mM EDTA, 50% Glycerol).

### Sample preparation for cryo-EM

The double stranded DNA substrates were prepared by mixing equimolar amounts of primer (Cryo3TP or CryoMM-2P Supplementary Table 2) and template (Cryo3TT or CryoMM-2T Supplementary Table 2) to a final concentration of 50 µM in dH$_2$O followed by heating to 95 °C for 10 min and allowed to cool to room temperature. The PolD-PCNA-DNA complex was associated by mixing 1 µM Pol D, 3 µM PCNA and 1.2 µM DNA in 20 mM HEPES pH8, 150 mM NaCl and 2 mM MgCl$_2$. The complex was incubated at RT for 20 min and pipetted onto glowed discharged cryo-EM grids (Quantifoil R 2/2 Au 300). Grids were frozen in liquid ethane by using a Vitrobot Mark IV (ThermoFischer) at 100% humidity, 20 °C, blotting force 0, waiting time 30 s, and blotting time 4 s.

### Cryo-EM data collection and image processing

Data collection was carried out at the Nanoimaging Core Facility at Pasteur Institute, Paris. Frozen grids were first screened using a 200 kV Glacios (Thermo Fischer) microscope to select optimal grids for cryo-EM data collection. Selected grids were imaged using a 300 kV Titan Krios (Thermo Fischer) microscope equipped with a K3 detector and a bio-quantum energy filter (Gatan). The defocus range, electron exposure, voltage, pixel size and other parameters of the data collection are present on Supplementary Table 1.

Image processing was carried out utilizing cryoSPARC[66] (version 4.2.1) and the workflow presented in Supplementary Figs. 1 and 2. A set of 500 random micrographs was used to optimize the particle picking strategy, and selected particles were used to train a Topaz[67] particle picking model. Topaz particle picking and extraction with all micrographs was performed followed by 2D classification to filter out bad picks. Ab-initio reconstruction and homogeneous refinement steps yielded a map of high global estimated resolution, but with visibly blurry density around the DP1 domain. After assessing multiple ways to deal with the heterogeneity of the complex with local refinement, 3D variability analysis and 3D classification, we found that 3D classification performed best to isolate populations of different PolD conformers. Non-uniform refinement[68] was carried out for each of the resulting classes. Two conformers were selected for the dataset containing DNA with three consecutive mismatches (Cryo 3 T, PDBid: 8PPU and 8PPV), one corresponding to the most open conformation of the claw and the other one, to the closest conformation; the closest conformer was selected for the dataset with a DNA substrate containing a single

mismatch (CryoMM-2, PDBid 8PPT). For both datasets, the particles from the selected classes were used to start a new 3D classification, as there was no increase in the resolution, or relevant differences on the map, the mother classes were kept. Local resolution estimation and particle orientation distribution plots of the final maps are shown in Supplementary Figs. 1 and 2.

## Building and refinement of cryo-EM model
PDB entry 6T8H[35] was used as a starting model and manually fit into the Cryo-EM maps in ChimeraX 1.6.[69,70] Notably, an AlphaFold2[71,72] prediction was used to improve or complete several regions of the model. An initial rigid body fit refinement was carried out using Phenix refine[73]. Then DP1 residues 173-195, 214-217, and DP2 residue 308 were manually modeled in Coot[74,75]. Additional real-space refinement was carried out in Coot, followed by real space refinement in Phenix. Further model refinement was carried out using Isolde[76] on ChimeraX. Multiple iterations of Phenix real space refinement and Isolde were carried out until convergence of refinement statistics was reached.

## Structural analysis
The surface interaction calculations were carried out using the PDBe-PISA tool[77]. UCSF ChimeraX[69,70] (version 1.6) was used for the generation of all figures containing cryo-EM maps and protein/DNA models.

## Capillary electrophoresis activity assays
To assess the exonuclease and polymerase activities of PolD variants, we performed capillary electrophoresis (CE) activity assays[78]. For CE visualization, primers with 5′-FAM fluorophore, and unlabeled templates were synthesized by, (Supplementary Table 2). P/T substrates were prepared by mixing 1 μM of 5′-FAM-labeled primer and 1.5 μM unlabeled template in annealing buffer (10 mM Tris-HCl, 100 mM NaCl, pH 7.5) and were incubated at 95 °C for 10 min and slowly cooled to room temperature. For primer excision and extension experiments, the FAM-labeled primer, MM-P was annealed to templates $MM^0T$ through $MM^{-5}T$ (Supplementary Table 2). For mismatch bypass experiments, FAM-labeled primers, $MB^0P$ through $MB^{-6}P$ were annealed to template MBT (Supplementary Table 2). Primer excision experiments were carried out by incubating 50 nM (final) of Pol D (or PolD mutants) with 5 nM (final) DNA substrate ($MM^0$-$MM^{-5}$) in Thermopol buffer (20 mM Tris-HCL, 10 mM $(NH4)_2SO_4$, 10 mM KCl, 2 mM $MgSO_4$, 0.1% Triton® X-100, pH 8.8) (B9004 New England Biolabs, Ipswich MA) in a total volume of 120 μL at 65 °C. 10 μL aliquots were removed at different time points (0.5, 1, 1.5, 2, 2.5, 3, 3.5 min) and quenched with equal volume of 50 mM EDTA and diluted with 30 μL $H_2O$. Experimental triplicates were performed for all reactions. DNA fragments were separated by capillary electrophoresis using a 3730xl Genetic Analyzer (Applied Biosystems) and fluorescent peaks were analyzed using Peak Scanner software version 1.0 (Applied Biosystems). Primer extension experiments were carried out as done for the primer excision tests with the addition of 100 μM (final) dNTPs. Mismatch bypass experiments were carried out by incubating 50 nM (final) of PolD, (or PolD mutants), 5 nM (final) DNA substrate and 20 μM (final) dNTPs, in Thermopol buffer at a final volume of 40 μL for 15 min at 65 °C. 10 μL of the reaction was quenched with equal volumes of 50 mM EDTA. To the remaining 30 μL, 4 units of BstBI (R0519 New England Biolabs Inc., Ipswich MA) was added, and further incubated at 65 °C for 15 min. 10 μL of the BstBI-reacted reaction was quenched with equal volumes of 50 mM EDTA. Experimental triplicates were performed for all reactions. All reactions were diluted with 30 μL $H_2O$ and analyzed by CE as described above.

## Pre-steady-state PolD exonuclease kinetics
To quantitate the observed rate constant of the 3′–5′ exonuclease activity of PolD (or PolD mutants), pre-steady-state, single-nucleotide excision experiments were performed. A 5′-FAM labeled primer with a single phosphorothioate bond present on the penultimate phospho-diester bond was synthesized (Kin-P) (Integrated DNA Technologies) along with two unlabeled templates, perfectly matched (Kin-FM) or containing a single terminal mismatch (Kin-MM-1). Primer / template substrates were prepared as described above. Equal volumes of substrate and PolD were mixed to reach a final concentration of 80 nM substrate and 200 nM PolD (or mutant) in 1 x Thermopol buffer (NEB) and incubated at 60 °C for various time points (see below). For $PolD^{wt}$ and PolD $R1178A^{DP2}$ and the single stranded DNA substrate (Kin-P), the assay was carried out using a rapid quench flow instrument (RQF) (KinTek Corp., Snow Shoe, PA) where the substrate and enzyme were rapidly mixed from 1 to 100 s and quenched with 500 mM EDTA. For the other mutants and substrates, the reactions were performed by manual mixing and incubated using a thermal cycler at 60 °C. The reactions were quenched after 15 to 105 s with of 500 nM EDTA. The reaction products were analyzed by capillary electrophoresis as described above. The time courses were collected in triplicate and the data were fitted to Eq. 1 to obtain the observed rate constant of single nucleotide excision ($k_{obs}$) using the non-linear regression program Kaleidagraph (Synergy Software).

$$[Product] = A[1 - e^{-k_{obs}t}] \tag{1}$$

Where A is the amplitude, t is time, and $k_{obs}$ is the observed rate constant.

## Reporting summary
Further information on research design is available in the Nature Portfolio Reporting Summary linked to this article.

## Data availability
The cryo-EM maps and all-atom models of the structures presented in this study have been deposited in the Protein Data Bank and EM data bank and are available with the following identifiers: Exo conformer of PolD containing a single mismatch with 8PPT and EMD-17815, exo conformer of PolD containing three consecutive mismatches 8PPU and EMD-17816 and intermediate conformer of PolD containing three consecutive mismatches 8PPV and EMD-17817. Source data are provided with this paper.

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

## Acknowledgements

We would like to thank Eric Beguec, as well as the leadership of New England Biolabs Inc. for fostering a supportive research environment. We wish to thank Dr. Clément Madru for helpful training and advise in sample preparation as well as figure making. We also thank Dr. Mart Krupovic for his help on the analysis of phylogenetic data. We would like to thank Danielle Fuchs, Harry Bell, Kristen Augelwicz, Dora Posfai and Laurie Mazzola for technical assistance. We would like to thank the Nanoimaging core facility members at Pasteur Institute for their assistance on the microscopes. We acknowledge the ESRF cryo-EM facility and the staff of CM-1 for their assistance during preliminarily data collection. We gratefully acknowledge the financial support from the ANR grant 'Archaprim' ANR-20-CE11-0003. L.B.A. was funded by a Cifre doctoral grant (ANRT). M.M.C. was funded by a postdoctoral FRM fellowship ('Fondation pour la Recherche Médicale'). We would also like to thank the CACSICE program for funding the cryo-EM equipment used in this study. Finally, we wish to thank Ahmed Haouz and Daniel Kneller for manuscript revision and discussion.

## Author contributions

L.B.A., K.M.Z., A.F.G., and L.S. conceived the experiments and analyzed the data. L.B.A. performed molecular cloning and protein purification. L.B.A. prepared the cryo-EM grids. L.B.A. and M.M.C. collected and analyzed the cryo-EM data. L.B.A. and L.S. reconstructed the cryo-EM map and analyzed the structure. L.B.A. and K.M.Z. conducted and analyzed all the activity assays.

## Competing interests

L.B.A., K.M.Z., and A.F.G. are employed and funded by New England Biolabs, Inc. and New England Biolabs France, manufacturers and vendors of molecular biology reagents, including DNA replication and repair enzymes. This affiliation does not affect the authors' impartiality, objectivity of data generation or its interpretation, adherence to journal standards and policies or availability of data. The remaining authors declare no competing interests.
