## [Peer Review File · Nature Communications]

Molecular basis for proofreading by the unique exonuclease domain of Family-D DNA polymerasesREVIEWER COMMENTS

Reviewer #1 (Remarks to the Author):

Betancurt-Anzola et al. present the first cryoEM structures of an editing complex of the archaeal family D polymerase polD. Several variants were designed, informed by the structure and tested for activity.

The work is of high quality, the manuscript is well written and the figures are clear for the most part. These new cryoEM structures shed light for the first time on the molecular mechanism of proofreading in this family of polymerases. The evolutionary aspect is fascinating, because PolD uses an accessory protein with a PDE fold for its exonuclease activity rather than a Dna-Q fold like other replicative polymerases.

I am very enthusiastic about this work and believe that it will be of wide interest to Nature Communications readers. I have a few minor comments below:

Page 4, description of excision and extension intermediates with mismatch substrates: What is the extension intermediate seen to the left of the fully extended product? It is relatively small compared to the full-length extension product with normal DNA but is comparable in size to the fully extended product in the context of some of the mismatches.

Page 6: The description of what Metal A does is a little vague, as a metal ion always "coordinates residues". It would be useful to specify which residues are coordinated.

Lines 197-199: include references

Page 7: Which program was used to calculate the interaction area between the primer strand and protein? The surface area numbers seem large to me.

Page 10: lines 337-338: "For example, a beta-hairpin originating from the exonuclease domain of family -A and -B DNA polymerases has been shown to play a key role in the melting of the primer and template strands".

Family A polymerases do not harbor the beta hairpin seen in family B polymerases (see ref #53). Also rephrase to say "most B DNA polymerases" because pol epsilon does not have an extended beta hairpin loop (PMID: 25550436).

Page 11: Intermediate conformers between pol and exo were also captured with RB69 (PMID: 15057283).

Page 13, lines 431-432: "the ancestral inactivated PDE domains act as a scaffold to anchor the contemporary eukaryotic polymerases to the replicative helicase." I don't understand this statement.

Material and Methods:

The expression vector adds a 42-residue tag to the N-terminus of DP1. These residues are resolved from the EM maps and thus ordered. I am just wondering where they lie relative to DP2.

The authors do not say why they chose an 18/25mer primer template for the structural studies. Similarly why did they choose 3x T and G:A mismatches?. It would be useful to get a sense as to how many DNA sequences they tried before obtaining a complex suitable for high-resolution cryo EM.

Figures:

General comment: yellow on white is generally hard to see because of the lack of contrast (e.g., DP1

label in Fig 2 a,c).

Figure 2F:

There is space to indicate the nature of each DNA base within the pyrimidine or purine symbol.

Figure 2H: I find it hard to see the difference between the two DNA models. It may be clearer without the protein residues.

Figure 6: Not clear if this putative evolutionary scenario is based on bona fide phylogenetic data?

Figure legends and elsewhere: Each PDBID code should be associated with a citation.

Supplemental information:

It would be informative to show a final purification gel of PolD and indicate which fraction was used for the EM studies.

How was AlphaFold2 used in the refinement of the models (supp Table 1)?

Reviewer #2 (Remarks to the Author):

Dear Leonardo and co-authors,

I have read your article entitled "Molecular basis for proofreading by the unique exonuclease domain of Family-D DNA polymerases" and found it to be a valuable contribution to the field.

The cryo-EM structures of the PolD core replisome and careful analysis provided in your work are particularly noteworthy. In addition, your main claims are well supported by the structures, their analysis, and the validation studies using mutants and different in-vitro approaches. I particularly appreciate how in-vitro experiments on mutants and WT proteins, based on initial structures, led to insights regarding primer strand partitioning and exonuclease mechanisms. Furthermore, the effort to analyse and discuss the evolutionary context and implications makes this work more appealing for a wider readership.

While your methodology is sound, I suggest providing more detailed information about your cryo-EM data processing procedures, especially model building. For example, there is no information regarding refinement parameters or validation. Please include details of the most important (not all!) parameters used during the different stages of map and pdb refinement so anyone can reproduce your work. Additionally, I noticed that all models have severe problems, particularly with clashes. Although these are likely still non-final models, addressing these issues before the deposition of maps and models and acceptance of this work is essential.

Overall, your work is sound and provides novel structures and insights that aid in understanding DNA polymerase proofreading mechanisms. Nevertheless, I have a few comments that should be addressed:

- As mentioned above, please add more details to the methods section regarding "Building and refinement of cryo-EM model".

- Figure 3d: I would recommend using mutant numbers for the double and triple mutants instead of just "double" and "triple". This would make the figure more straightforward to follow without resorting to text.

- Line 273-275 & Figure 3d-e: the text states that reduction in activity for mutant R1178A with P/T substrates is ~60%, which matches the info in the table (Fig3e) for the fully paired substrate, but it doesn't for the mismatched substrate (97% reduction). By looking at the graphs in Fig3d, it seems that there might be an error somewhere when getting the Kobs for the mismatched substrate. Please review and correct accordingly.

- Fig3 & Fig4: Why haven't mutants P413A, P1107A and R1114A been included in the steady state experiments? If data is available, it would be advisable to include it for completeness of the analysis.

- Supplementary figures 1 & 2: please include model-vs-map FSC.

- Supplementary table 1:

o Please review models and update (clash score particularly bad).

o Ramachandran plot values for the third model are inverted, please correct.

o Model resolution shouldn't be better (lower number) than map resolution. Please review and/or comment.

We would like to take the opportunity to sincerely thank the reviewers for their enthusiastic comments about our work, and for the important points that they raised in order to help us improving the manuscript. You will see that we have considered the reviewers' comments very seriously and done our best to answer all of them.

Point-by-point answers to the reviewer's comments

Reviewer #1 (Remarks to the Author):

Betancurt-Anzola et al. present the first cryoEM structures of an editing complex of the archaeal family D polymerase polD. Several variants were designed, informed by the structure and tested for activity.

The work is of high quality, the manuscript is well written and the figures are clear for the most part. These new cryoEM structures shed light for the first time on the molecular mechanism of proofreading in this family of polymerases. The evolutionary aspect is fascinating, because PolD uses an accessory protein with a PDE fold for its exonuclease activity rather than a Dna-Q fold like other replicative polymerases.

I am very enthusiastic about this work and believe that it will be of wide interest to Nature Communications readers. I have a few minor comments below:

Page 4, description of excision and extension intermediates with mismatch substrates: What is the extension intermediate seen to the left of the fully extended product? It is relatively small compared to the full-length extension product with normal DNA but is comparable in size to the fully extended product in the context of some of the mismatches.

The peak to the left of the fully extended product is a consequence of the polymerase almost reaching the end of the substrate and pausing. We have shown that PolD from *P.abysyi* binds to DNA primer/template overhangs with a much higher affinity compared to double stranded DNA with blunt ends (PMID 30657780). Substrates with very short template overhangs, or blunt ends, are not optimal substrates for PolD. Notably, after longer incubation times, the polymerase extends all the substrate to full length product and the peak disappears, as shown in the traces below.

We added a sentence to the legend of Figure 1 to explain the presence of this peak: ‘Notably extension pausing is observed at short timepoints when the polymerase reaches the end of the template.’

Page 6: The description of what Metal A does is a little vague, as a metal ion always “coordinates residues”. It would be useful to specify which residues are coordinated.

Thank you for the observation, we have added the residues that coordinate metal ion A to the manuscript. The sentence, found in lines 169-171, now reads: “Mg²⁺-A has been shown to coordinate residues Q954, D956, and D958 in the polymerase active site, while Mg²⁺-B, which is not observed in these structures, is supplied by the incoming nucleotide.”

Lines 197-199: include references

The following references have been added accordingly: PMID: 28067916, PMID: 8469987, PMID: 22902479, PMID: 37202477, PMID: 10535734

Page 7: Which program was used to calculate the interaction area between the primer strand and protein? The surface area numbers seem large to me.

For the surface calculation we used two different tools, the PDBePISA web server, and AreaMol (CCP4 suite). We obtained similar values with these two tools. The figure below shows an example of the PDBePISA result with the last 4 bases of the primer and PolD (As shown in the figure on the bottom). Lines 5 and 7 show the interactions between the primer and DP1 and DP2 respectively.

PISA Interface List.

Session Map (id=974-H6-4K8)
 Start Warnings Interfaces Interface Search
 Monomers
 Assemblies

Interfaces in PDBePISA_Test.pdb crystal.

Space symmetry group: P 1

##		Structure 1			x	Structure 2			Interface		ΔG	ΔG	N_{iB}	N_{iS}	N_{iD}	CSS		
Id	NN	Range	N_{iB}	N_{iS}	Surface \AA^2	Range	Symmetry op-n	Sym.ID	N_{iB}	N_{iS}	Surface \AA^2	area, \AA^2	kcal/mol	P-value				
1	1	B	205	48	50705	∩ A	x,y,z	1_555	229	67	21206	1981.5	-24.4	0.016	10	4	0	0.325
2	2	E	62	15	12999	∩ C	x,y,z	1_555	64	19	13157	604.4	-4.3	0.292	8	7	0	0.000
3	3	E	69	18	12999	∩ D	x,y,z	1_555	64	15	13204	595.1	-4.0	0.309	6	8	0	0.000
4	4	D	56	18	13204	∩ C	x,y,z	1_555	58	14	13157	563.1	-5.6	0.165	4	2	0	0.000
Average:												587.5	-4.7	0.255	6	6	0	0.000
3	5	P	60	4	1336	∩ A	x,y,z	1_555	71	21	21206	562.4	-2.9	0.158	0	0	0	0.122
4	6	B	48	9	50705	∩ C	x,y,z	1_555	61	17	13157	499.3	-5.4	0.090	6	0	0	0.000
5	7	P	36	3	1336	∩ B	x,y,z	1_555	38	11	50705	338.3	-0.5	0.651	4	0	0	0.022
6	8	B	35	10	50705	∩ D	x,y,z	1_555	34	9	13204	308.0	0.7	0.504	5	4	0	0.000
7	9	C	26	9	13157	∩ A	x,y,z	1_555	18	6	21206	167.1	-1.0	0.466	0	0	0	0.000
8	10	B	11	5	50705	∩ E	x,y,z	1_555	9	4	12999	79.9	-0.0	0.580	0	0	0	0.000
9	11	[ZNF:1302]	1	1	98	f B	x,y,z	1_555	3	3	50705	49.2	-39.0	0.000	0	0	0	0.430
10	12	[MG]F:1	1	1	98	f B	x,y,z	1_555	7	3	50705	45.5	-6.5	0.000	0	0	0	0.072
11	13	[MG]F:3	1	1	98	f A	x,y,z	1_555	11	6	21206	37.6	-6.1	0.000	0	0	0	0.514
14	14	[MG]F:2	1	1	98	f A	x,y,z	1_555	5	4	21206	35.1	-6.0	0.000	0	0	0	0.514
Average:												36.4	-6.1	0.000	0	0	0	0.514
12	15	[MG]F:3	1	1	98	∩ P	x,y,z	1_555	4	1	1336	25.0	-3.2	0.000	0	0	0	0.217
16	16	[MG]F:2	1	1	98	∩ P	x,y,z	1_555	4	1	1336	14.9	-1.9	0.000	0	0	0	0.217
Average:												20.0	-2.6	0.000	0	0	0	0.217
13	17	[MG]F:3	1	1	98	f [MG]F:2	x,y,z	1_555	1	1	98	17.4	-3.5	0.000	0	0	0	0.147
14	18	E	2	1	12999	∩ A	x,y,z	1_555	1	1	21206	2.5	0.1	0.735	0	0	0	0.000

Notably, the most important results derived from these surface calculations are the comparisons between different DNA polymerases, found in lines 208 – 211 and 358 - 360 of the manuscript, rather than their absolute values.

Page 10: lines 337-338: “For example, a beta-hairpin originating from the exonuclease domain of family -A and -B DNA polymerases has been shown to play a key role in the melting of the primer and template strands”. Family A polymerases do not harbor the beta hairpin seen in family B polymerases (see ref #53). Also rephrase to say “most B DNA polymerases” because pol epsilon does not have an extended beta hairpin loop (PMID: 25550436).

We agree with the reviewer, the text has been modified accordingly, and the reference added. The sentence, found in lines 340 -342, now reads: ‘For example, a beta-hairpin originating from the exonuclease domain of most family -B DNA polymerases has been shown to play a key

role in the melting of the primer and template strands,'

Page 11: Intermediate conformers between pol and exo were also captured with RB69 (PMID: 15057283).

We thank the reviewer for presenting this information to us. We have added a sentence to illustrate that RB69 DNA polymerase was also captured in different conformations. This sentence, found in lines 386 – 388 reads: “Additionally, RB69 DNA polymerase crystallized with DNA primer with a template containing an abasic site, shows four different conformations in both exo and pol modes” (PMID: 15057283)

Additionally, we calculated the distance between RB69 exonuclease and polymerase active sites using the PDBePISA web server, and added this distance, 37 Å, to line 359 of the manuscript to provide an additional data point for the calculated distance between various DNA polymerase exonuclease and polymerase active sites.

Page 13, lines 431-432: “the ancestral inactivated PDE domains act as a scaffold to anchor the contemporary eukaryotic polymerases to the replicative helicase.” I don't understand this statement.

Thank you for this comment, we have added additional context to that paragraph to be clearer. These sentences, found in lines 439-444 now reads:

“Indeed, while the architecture of the PDE domain of PolD has been conserved in the B-subunit of all three eukaryotic repDNAPs (Pol α , Pol δ and Pol ϵ), its nuclease activity has been lost over the course of evolution. In eukaryotes, the ancestral polymerase-associated PDE has thus been replaced by a DnaQ nuclease fold in all repDNAPs. Importantly DP1 and the B-subunit of eukaryal repDNA polymerases share a role in recruiting different replication factors such as GINS, Primase, or even MCM.”

Material and Methods:
The expression vector adds a 42-residue tag to the N-terminus of DP1. These residues are resolved from the EM maps and thus ordered. I am just wondering where they lie relative to DP2.

We apologize for the confusion surrounding the N-terminus of DP1. In fact, the N-terminal region of DP1 is not defined in the map, and the N-terminal expression tag as well as residues 1-172 of DP1 are not ordered and thus not modeled. We added the following sentence, found in lines 551 – 553 of the methods section: “The N-terminal region of DP1 (1-172) and the N-terminal expression tag were not defined in the map and were not included in the model.”

Interestingly, this region contains a domain that interacts with the CMG helicase (PMID: 34568951) and is flexible in the absence of this interaction.

The authors do not say why they chose an 18/25mer primer template for the structural studies. Similarly why did they choose 3x T and G:A mismatches?. It would be useful to get a sense as to

how many DNA sequences they tried before obtaining a complex suitable for high-resolution cryo EM.

In previous work, an 18/25mer well-matched double stranded DNA substrate was optimized to capture a stable ternary complex (PolD-PCNA-DNA) (PMID: 32221299). To capture PolD in a proofreading mode, we used this 18/25mer substrate as a scaffold and altered it with two different strategies:

First, we designed a substrate that mimics an open fork. We screened two lengths of the primer: three and five consecutive T mismatches. The substrate with five consecutive mismatches produced a structure with the primer absent from the exonuclease domain (Based on our structures, we showed later that the DP1 exo channel only accommodates 4 nucleotides). However, the three consecutive mismatch substrate yielded the two Cryo-EM structures (8PPU and 8PPV) with the 3T primer clearly entering the DP1 exonuclease domain.

Second, we designed substrates containing one single mismatch at different positions (shown in red in the figure below). We carried out one primer extension screening experiment to determine the position of the mismatch that prevented nucleotide incorporation and potentially stalled the polymerase in the exonuclease mode. We observed that a mismatch at position -2 (red box in figure below) stalls the polymerase and prevents it from extending a primer. This substrate yielded the third CryoEM structure (8PPT)

Figures:

General comment: yellow on white is generally hard to see because of the lack of contrast (e.g., DP1 label in Fig 2 a,c).

We agree with the reviewer: The color was changed to a darker yellow, see Fig 2.

Figure

2F:

There is space to indicate the nature of each DNA base within the pyrimidine or purine symbol.

We agree with the reviewer. The nature of each base was added inside the symbol, making the figure clearer. See figure 2F.

Figure 2H: I find it hard to see the difference between the two DNA models. It may be clearer without the protein residues.

We agree with the reviewer and therefore we removed the protein residues from Figure 2H. The difference between the two DNA molecules is now more visible. Thank you for the suggestion.

Figure 6: Not clear if this putative evolutionary scenario is based on bona fide phylogenetic data?

We agree with the reviewer that this is an important question. Indeed, the tree we proposed is supported by phylogenetic data (see below). The following tree shows how Mre11, DP1, and the regulatory B subunits of replicative eukaryal DNA polymerases are grouped.

Importantly, this tree is also supported by a more thorough phylogenetic analysis of phosphoesterase motifs, that we cite in this manuscript (PMID: 9685491). Lines 437-441

Figure legends and elsewhere: Each PDBID code should be associated with a citation.

We agree with the reviewer. The citations were added after each PDBid

Supplemental information:
It would be informative to show a final purification gel of PolD and indicate which fraction was used for the EM studies.

Below, we show the size exclusion column chromatogram and SDS-PAGE gel for the final purification of PolD that was used for EM studies. Following the reviewer's suggestion, we created a new Supplementary Figure (Supplementary Figure 8) that contains this information.

How was Alphafold2 used in the refinement of the models (supp Table 1)?

The Alphafold2 prediction of DP1/DP2 was used to generate the initial .eff file for structural refinement. In fact, the alpha fold model revealed some extra short beta strands and alpha helices, in comparison to our previous Cryo-EM structures of PolD (PDBID 6T8H). Additionally, we were able to model some extra residues (notably in DP1) by using the AF model.

Reviewer #2 (Remarks to the Author):

Dear Leonardo and co-authors,

I have read your article entitled "Molecular basis for proofreading by the unique exonuclease domain of Family-D DNA polymerases" and found it to be a valuable contribution to the field. The cryo-EM structures of the PolD core replisome and careful analysis provided in your work are particularly noteworthy. In addition, your main claims are well supported by the structures, their analysis, and the validation studies using mutants and different in-vitro approaches. I particularly appreciate how in-vitro experiments on mutants and WT proteins, based on initial structures, led to insights regarding primer strand partitioning and exonuclease mechanisms. Furthermore, the effort to analyse and discuss the evolutionary context and implications makes this work more appealing for a wider readership.

While your methodology is sound, I suggest providing more detailed information about your cryo-EM data processing procedures, especially model building. For example, there is no

information regarding refinement parameters or validation. Please include details of the most important (not all!) parameters used during the different stages of map and pdb refinement so anyone can reproduce your work.

We agree with the reviewer. A more exhaustive description of the map refinement strategy has been added to the manuscript:

Image processing was carried out utilizing cryoSPARC (version 4.2.1) (PMID: 28165473) and the workflow presented on **Supplementary Fig. 1** and **2**. A set of 500 random micrographs was used to optimize the particle picking strategy, and selected particles were used to train a Topaz (PMID: 31591578) particle picking model. Topaz particle picking and extraction with all micrographs was performed followed by 2D classification to filter out bad picks. *Ab-initio* reconstruction and homogeneous refinement steps yielded a map of high global estimated resolution, but with visibly blurry density around the DP1 domain. After assessing multiple ways to deal with the heterogeneity of the complex with local refinement, 3D variability analysis and 3D classification, we found that 3D classification with a focus mask over DP1 performed best to isolate populations of different PolD conformers. Non-uniform refinement was carried out for each of the resulting classes. Two conformers were selected for the dataset containing DNA with three consecutive mismatches (Cryo 3T, PDBid:8PPU and PDBid:8PPV), one corresponding to the most open conformation of the claw and the other one to the most closed conformation. The closed conformer was also selected for the dataset with a DNA substrate containing a single mismatch (CryoMM-2, 8PPT). Local resolution estimation and particle orientation distribution plots of the final maps are shown in **Supplementary Fig. 1** and **2**.

Additionally, I noticed that all models have severe problems, particularly with clashes. Although these are likely still non-final models, addressing these issues before the deposition of maps and models and acceptance of this work is essential.

Overall, your work is sound and provides novel structures and insights that aid in understanding DNA polymerase proofreading mechanisms.

Nevertheless, I have a few comments that should be addressed:
- As mentioned above, please add more details to the methods section regarding “Building and refinement of cryo-EM model”.

We agree with the reviewer. A more exhaustive description of the refinement strategy has been added to the manuscript:

‘PDB entry 6T8H was used as a starting model and manually fit into the cryo-EM maps in ChimeraX (PMID: 3288110, PMID: 28710774). Notably, an AlphaFold2 prediction (PMID: 35637307, PMID: 34265844) was used to improve or complete several regions of the model. An initial rigid body fit refinement was carried out using Phenix refine (PMID: 21821126), then DP1 residues 173-195, 214-217 and DP2 residue 308 were manually modelled

in Coot (PMID: 15572765, PMID: 20383002). Additional real space refinement was carried out in Coot, followed by real space refinement in Phenix. Further model refinement was carried out using Isolde on ChimeraX (PMID: 29872003). Multiple iterations of Phenix real space refinement and Isolde were carried out until convergence of refinement statistics was reached.'

- Figure 3d: I would recommend using mutant numbers for the double and triple mutants instead of just “double” and “triple”. This would make the figure more straightforward to follow without resorting to text.

We agree with the reviewer. The figures 4 and 5, and the manuscript were modified accordingly.

- Line 273-275 & Figure 3d-e: the text states that reduction in activity for mutant R1178A with P/T substrates is ~60%, which matches the info in the table (Fig3e) for the fully paired substrate, but it doesn't for the mismatched substrate (97% reduction). By looking at the graphs in Fig3d, it seems that there might be an error somewhere when getting the Kobs for the mismatched substrate. Please review and correct accordingly.

We sincerely would like to thank the reviewer for noticing this error. There was indeed a problem when reporting the data on the table, the values were corrected accordingly in Figure 3e, and are highlighted below for clarity.

Table before correction

e

Mutant	$K_{obs} \text{ (sec}^{-1}) \times 10^{-2} \text{ }^a$			Relative K_{obs} to wild type (%)		
	ssDNA	P/T DNA _{fully paired}	P/T DNA _{mismatch-1}	ssDNA	P/T DNA _{fully paired}	P/T DNA _{mismatch-1}
wt	8.1 ± 1.1 ^b	2.3 ± 0.3	5.3 ± 0.1	100	100	100
Y412A	3.9 ± 0.4	0.5 ± 0.1	1 ± 0.02	48	22	19
F586A	3.4 ± 0.3	0.4 ± 0.03	0.7 ± 0.1	42	17	13
R499A	1.4 ± 0.7	0.12 ± 0.007	0.22 ± 0.01	17	5	4
Double	0.8 ± 0.2	0.03 ± 0.003	0.07 ± 0.01	10	1	1
R1178A	7.6 ± 0.8 ^b	0.9 ± 0.2	0.12 ± 0.01	94	40	2.3
Triple	~0	~0	~0	~0	~0	~0

Table after correction

e

Mutant	$K_{obs} \text{ (sec}^{-1}) \times 10^{-2} \text{ }^a$			Relative K_{obs} to wild type (%)		
	ssDNA	P/T DNA _{fully paired}	P/T DNA _{mismatch-1}	ssDNA	P/T DNA _{fully paired}	P/T DNA _{mismatch-1}
wt	8.1 ± 1.1 ^b	2.3 ± 0.3	5.3 ± 0.1	100	100	100
Y412A	3.9 ± 0.4	0.5 ± 0.1	1 ± 0.02	48	22	19
F586A	3.4 ± 0.3	0.4 ± 0.03	0.7 ± 0.1	42	17	13
R499A	1.4 ± 0.7	0.12 ± 0.007	0.22 ± 0.01	17	5	4
Y412A/F586A	0.8 ± 0.2	0.03 ± 0.003	0.07 ± 0.01	10	1	1
R1178A	7.6 ± 0.8 ^b	0.9 ± 0.2	1.2 ± 0.01	94	40	23
Y412A/F586A/R499A	~0	~0	~0	~0	~0	~0

- Fig3 & Fig4: Why haven't mutants P413A, P1107A and R1114A been included in the steady state experiments? If data is available, it would be advisable to include it for completeness of the analysis.

Through mismatch bypass experiments (**Figure 5**) and qualitative exonuclease assays (shown below and in the **new Supplementary Figure 7**) we observed minimal effect on PolD exonuclease activity with P413A, P1107A and R1114A mutants. We therefore decided to prioritize mutants that did effect PolD exonuclease activity in our kinetic experiments. We have added a sentence to Results (lines 261 – 264) to provide clarity to the manuscript. These sentence reads: Note that mutants P413A^{DP1}, P1107A^{DP2}, and R1114A^{DP2} were not included in this study as their excision profiles do not show a strong effect on the exonuclease activity (**Supplementary Fig. 7**) and mismatch bypass (See below: Mismatch bypass of exo channel PolD mutants and **Figure 5**).

- Supplementary figures 1 & 2: please include model-vs-map FSC.

We agree with the reviewer, the model-vs-map FSCs were added for the three models to Supplementary Figures 1&2.

8PPT (Model vs map FSC)

8PPU Model vs map FSC

8PPV model vs map FSC

- Supplementary table 1:

o Please review models and update (clash score particularly bad).

Following the reviewer's suggestion, we improved our models, in particular the bad clash score was improved using Isolde, coot and phenix. These new models are on hold by the PDB, the new validation reports didn't show major issues. We also added extra information in materials and methods to mention the usage of Isolde (found on lines 554-555). Below we added the global validation metrics before (left) and after (right) revision.

8PPT

8PPU

8PPV

o Ramachandran plot values for the third model are inverted, please correct.

Thank you for noticing this, we have corrected the values that were inverted in table S1

o Model resolution shouldn't be better (lower number) than map resolution. Please review and/or comment.

Thank you for this observation, the value was corrected. Below you can find a screenshot of the table before and after correction, with the correct highlighted below for clarity.

Previous table

Map resolution (Å)	3.02	3.02	2.90
FSC threshold	0.143	0.143	0.143
Map resolution range (Å)	2.5 - 4.5	2.5 - 4.5	2.5 - 4.5
Refinement			
Initial models used	PDB: 6T8H, AlphaFold2	PDB: 6T8H, AlphaFold2	PDB: 6T8H, AlphaFold2
Model resolution	2.99	3.00	2.87

Table with values corrected

Map resolution (Å)	3.02	3.02	2.90
FSC threshold	0.143	0.143	0.143
Map resolution range (Å)	2.5 - 4.5	2.5 - 4.5	2.5 - 4.5
Refinement			
Initial models used	PDB: 6T8H, AlphaFold2	PDB: 6T8H, AlphaFold2	PDB: 6T8H, AlphaFold2
Model resolution	3.3	3.3	3.2

REVIEWERS' COMMENTS

Reviewer #1 (Remarks to the Author):

The authors have adequately addressed all of my concerns. Actually I commend them for being thorough and thoughtful in their replies. Well done.

Reviewer #2 (Remarks to the Author):

I believe the authors have addressed all the comments adequately and that the manuscript is clear now. Figures and method information have improved as well during the revision. In my opinion, this should be ready for publication now.

REVIEWERS' COMMENTS

Reviewer #1 (Remarks to the Author):

The authors have adequately addressed all of my concerns. Actually I commend them for being thorough and thoughtful in their replies. Well done.

Reviewer #2 (Remarks to the Author):

I believe the authors have addressed all the comments adequately and that the manuscript is clear now. Figures and method information have improved as well during the revision. In my opinion, this should be ready for publication now.

We would like to thank again both reviewers for their work. Their useful advices helped us improving the manuscript.